# Therapeutic failure of multidrug therapy for leprosy: A retrospective case series in a hyperendemic Brazilian City

**Andrea Maia Fernandes de Araújo Fonseca**[1]*, **Patrícia Sammarco Rosa**[2],
**Andrea de Faria Fernandes Belone**[2], **Cleverson Teixeira Soares**[2],
**Daniele de Faria Ferreira Bertoluci**[2], **Suzana Madeira Diório**[2], **Luciana Raquel Vincenzi Fachin**[2],
**Rodrigo Feliciano do Carmo**[1], **Francisco Bezerra de Almeida Neto**[3]

**1** Federal University of the São Francisco Valley, Postgraduate program in Health and Biological Sciences, Petrolina, Pernambuco, Brazil, **2** Division of Research and Education, Lauro de Souza Lima Institute, Bauru, São Paulo, Brazil, **3** Department of Dermatology, Medicine, Maurício de Nassau University Center, Recife, Pernambuco, Brazil

* andrea.mfa@me.com

## Abstract

### Background

Leprosy remains endemic in many regions despite the global rollout of multidrug therapy (MDT). Clinical cure—defined by completion of a time-based MDT regimen—may not reflect proper bacteriological clearance, particularly in patients with persistent reactions or neurological symptoms. We aimed to assess subclinical disease activity in multibacillary patients who completed an extended 24-dose MDT course.

### Methods

In this retrospective case series, between January 2016 and November 2023, 131 multibacillary patients treated at the Petrolina Infectious Diseases Service (SEINPE) underwent skin biopsy upon completing 24 monthly MDT doses. Disease activity was evaluated by histopathology (H&E and Fite–Faraco staining; n = 123), slit-skin smear with bacilloscopic and morphological indices (BI, n = 126; MI, n = 74), qPCR for *M. leprae* (n = 101), and nude mice footpad inoculation (n = 45) at Instituto Lauro de Souza Lima, Bauru, Brazil. Drug-resistance mutations were detected by sequencing (*folP1, rpoB, gyrA*; n = 88). Neurological function was assessed using a Simplified Neurological Assessment (n = 117).

### Results

Histopathology revealed active disease or bacillary persistence in 62/123 specimens (50.41%), while 29/45 inoculations (64.44%) yielded viable bacilli. qPCR detected *M. leprae* DNA in 96/101 patients (95.05%). Known resistance mutations were identified in 2/88

**Data availability statement:** All relevant data are within the manuscript and its supporting information files.

**Funding:** This study was financed in part by the Coordination for the Improvement of Higher Education Personnel – Brazil (CAPES) – Process number 88881.708019/2022-01 - PDPG-CONSOLIDAÇÃO-3-4. The funders had no role in study design, data collection and analysis, decision to publish, or preparation of the manuscript.

**Competing interests:** The authors have declared that no competing interests exist.

patients (2.27%). Clinically, 89/131 patients (67.94%) no longer exhibited skin lesions post-MDT; however, neurological impairment increased from 70/131 (53.44%) at diagnosis to 114/131 (87.02%) at discharge. The proportion with grade 2 disability increased from 5/100 (5.00%) to 27/117 (23.08%). Exact 95% CIs are reported in the manuscript.

## Conclusions

More than half of patients treated with an extended 24-dose MDT regimen harbored persistent *M. leprae* activity despite apparent dermatological cure, and most experienced worsening neural function. Time-based discharge criteria alone are inadequate to confirm cure. We recommend integrating post-treatment histopathological, molecular, and inoculation assessments—particularly in patients with persistent reactions or neurological complaints—to identify therapeutic failure, guide retreatment, and prevent long-term disability.

## Author summary

Leprosy is a chronic infection of the skin and peripheral nerves caused by *Mycobacterium leprae* and its related species, *M. lepromatosis*. Untreated, it can lead to permanent nerve damage and disability. Globally, patients are considered "cured" after completing the World Health Organization's multidrug therapy (MDT) regimen, yet many continue to experience inflammation.

Between 2016 and 2023, we evaluated 131 multibacillary patients who received twice the standard duration of MDT (24 monthly doses). At treatment completion, histopathology revealed persistent bacilli or inflammation in half of the biopsies (50%), in addition, two-thirds of the mice inoculations (64%) confirmed the presence of viable bacteria.

Molecular testing detected resistance mutations in only two patients (2.3%), suggesting that drug resistance does not account for the majority of treatment failures. Clinically, 68% of patients had no visible skin lesions at discharge; however, 87% presented with persistent or worsening nerve problems. The proportion with severe disability (grade 2) increased from 5% at diagnosis to 23% after treatment.

These findings show that time-based treatment alone may not clear hidden *M. leprae* or prevent disability. We recommend post-treatment follow-up with tissue-based and biological tests to detect persistence, guide retreatment, and reduce long-term nerve damage and stigma.

## Introduction

Leprosy is a chronic infectious disease caused by *Mycobacterium leprae (M. leprae)* and *M. lepromatosis*, with a predilection for Schwann cells of the peripheral nervous

system and cutaneous tissues. Initial manifestations often involve neuropathic symptoms—pain, burning, stabbing sensations, shock-like paresthesia—predominantly in the hands and feet. Classical skin lesions with altered or decreased sensation typically follow these neural complaints, reflecting bacillary invasion of cutaneous nerves [1–3].

According to the January 2025 National Epidemiological Bulletin, Brazil reported 22,773 new leprosy cases in 2023, with the states of Maranhão (6.5/100,000), Mato Grosso (18.0/100,000), and Pernambuco (11.7/100,000) exhibiting the highest detection rates. In Petrolina (Pernambuco), 288 cases were reported, of which 273 (94.8%) were multibacillary and five occurred in children under 15 years [4,5]. Despite these substantial burdens, time-based treatment completion remains the sole criterion for declaring "cure".

Since its adoption in Brazil in 1982, the WHO-recommended multidrug therapy (MDT)—comprising rifampicin, dapsone, and clofazimine—has been the cornerstone of leprosy control. Paucibacillary cases receive six months of treatment, while multibacillary cases receive 12 months of treatment. Initially, guidelines advised continuing MDT until smear negativity or for a minimum of two years; the current 12-month limit reflects a shift in paradigm toward programmatic simplicity [6,7]. "Cure" is assumed upon regimen completion, without routine bacteriological or histopathological confirmation.

Nonetheless, acute inflammatory episodes (type 1 and type 2 reactions) frequently interrupt clinical stability, causing substantial morbidity, psychosocial impact, and economic burden. Management relies on prolonged corticosteroid therapy, which carries risks of severe adverse effects and complicates long-term care [8,9]. Moreover, silent neuritis—subclinical nerve inflammation detectable only through disability grading or biopsy—has been documented up to five years post-treatment, underscoring the immunological underpinnings of neural damage [10,11].

Although genetic mutations in *M. leprae* (*folP1*, *rpoB*, *gyrA*) explain dapsone, rifampicin, and ofloxacin resistance, such mechanisms account for a minority of treatment failures [12–14]. In Brazil, official documents allowed MDT extension to 24 doses for selected multibacillary cases until 2022, when new guidelines curtailed extension to all patients [12–17].

Although patients may receive regular MDT, undergo systematic household contact screening, and show no evidence of resistance in standard molecular assays, a substantial proportion still fails to achieve satisfactory clinical improvement. This observation suggests that additional pathogenic factors are likely involved. Among them, bacillary dormancy within macrophages and Schwann cells—promoted by lipid droplet biogenesis—may enable long-term survival of *M. leprae*. Furthermore, the contribution of as-yet uncharacterized drug resistance mechanisms cannot be excluded, potentially accounting for persistent disease despite therapy. [18].

Pharmacokinetic interactions among MDT components can also influence drug exposure. For instance, the co-administration of rifampicin and dapsone alters each other's metabolism, and individual acetylator genotypes may modulate dapsone efficacy, although clinically significant effects remain under investigation [19]. Histopathologically, bacillary regression is uneven: fragmented bacilli ("bacillary dust") are apparent by six months but may persist for years in macrophages; endothelial cells clear bacilli more rapidly than nerve sheaths, creating reservoirs of viable organisms [20]. Nude mice footpad inoculation remains the gold standard for confirming active infection in such cases [21,22].

Given this complex backdrop, critical gaps remain. To the best of our knowledge, no study has comprehensively assessed clinical signs, simplified neurological evaluation, histopathology, molecular detection, and *in vivo* inoculation in parallel following an extended 24-dose MDT regimen. Understanding the true extent of bacillary persistence—and its relationship to treatment failure—is essential for refining cure criteria and preventing irreversible neural sequelae.

Thus, this study aims to characterize therapeutic failure and assess clinical and subclinical *M. leprae* activity in multibacillary patients who completed an extended 24-dose MDT course, to re-evaluate the adequacy of time-based discharge criteria.

## Methodology

### Ethics statement

This study complied with Brazilian ethical requirements established by National Health Council Resolutions 466/2012 (for human participants) and 510/2016 (for secondary data). It was approved through the Ethics Committee of the University

Hospital of the Federal University of Vale do São Francisco (HU/UNIVASF) under Consolidated Opinion No. 6.195.103. The requirement for informed consent was waived because this was a retrospective case-series review of anonymized medical records with no direct patient interaction or additional risk. Therefore, all individuals remained anonymous and were over 18 years of age. Because the study involved live-animal procedures, specifically the intradermal inoculation of biopsy-derived suspensions into the hind footpads of athymic mice to assess the viability of *M. leprae*, animal use was reviewed and approved by the Lauro de Souza Lima Institute Ethics Committee on Animal Use (CEUA), protocol CEUA/ILSL 003/21. Animals were handled under anesthesia, pain and distress were minimized, and euthanasia was performed humanely at the predefined endpoint to allow terminal harvest of the inoculated footpads for bacillary recovery and viability assessment, under CEUA-approved procedures and national guidance. All data were handled per professional confidentiality standards, and no patient was identified at any stage of the study.

## Study design and setting

This is a retrospective census case-series of all eligible multibacillary patients re-evaluated after completing 24-dose MDT at referral services in Brazil (January 2016 and November 2023). Clinical care and data abstraction at the post-MDT index visit were conducted at the Petrolina Infectious Diseases Service (Pernambuco, Brazil). Laboratory assessments were performed at Lauro de Souza Lima Institute, Bauru, SP, including histopathology, nude mice inoculation, qPCR for *M. leprae*, and drug-resistance sequencing (*folP1, rpoB, gyrA*).

Reporting adheres to STROBE for observational studies; we also incorporate selected CARE elements adapted to a case-series (aggregate patient information and a cohort-level clinical timeline). Completed STROBE and CARE checklists with item-to-section cross-references are provided in the Supporting Information (S1 STROBE Checklist and S1 Care Checklist).

## Participants and eligibility criteria

All multibacillary patients (n = 131) evaluated at the post-MDT index visit during the study period were included if they met the prespecified criteria. Inclusion criteria required the presence of at least one of the following findings: (1) erythematous or infiltrated skin lesions observed after completion of MDT, in comparison to the clinical presentation at diagnosis; (2) persistence, worsening, or new onset of neurological symptoms—such as muscle cramps, paresthesias, or numbness in the extremities (hands and feet)—accompanied by acute or chronic pain, burning sensations, or spontaneous sensory disturbances; (3) deterioration of neural function documented by the Simplified Neurological Assessment (SNA), compared to findings at diagnosis; (4) persistent leprosy reactions, including type 1 or type 2 reaction episodes, or both, with or without neuritis, occurring separately or simultaneously, unresponsive to conventional treatment during or after MDT; these reactional episodes were documented either at discharge or within six months following treatment completion and coincided with the time of skin biopsy. No additional exclusion criteria were specified beyond incomplete records, which precluded assessment at the index visit.

## Variables and outcomes

Primary outcomes included evidence of persistent disease after MDT, as determined by histopathology, nude mice inoculation, and qPCR. Ancillary outcomes included drug-resistance genotyping (*folP1, rpoB, gyrA*) and disability grade by the Simplified Neurological Assessment (SNA). Because specific tests were performed in analytic subsets, all estimates were reported with explicit denominators; operational definitions and analytic denominators were consolidated in Table 1. Exact 95% confidence intervals for key proportions are reported in the main text.

## Data sources and measurements (overview)

Clinical data were abstracted from standardized charts at the index re-evaluation. Laboratory procedures followed institutional protocols. Detailed methods for defining MDT-therapeutic failure, collecting biological material, performing histopathology, inoculating nude mice, conducting qPCR, and sequencing for drug resistance are provided verbatim in the

**Table 1. Tests performed at the post-MDT index re-evaluation: operational definitions and analytic denominators.**

| Domain/Test | Specimen & method | Operational definition/ threshold | n with a valid result (denominator for analysis) | Notes (limits/ where detailed) |
|---|---|---|---|---|
| Slit-skin smear (BI, post-MDT) | Smear from standard sites | Ridley scale 0–6+ | 126/131 (96.2%) | Summary in Results |
| Morphological Index (post-MDT) | Smear from standard sites | % intact vs. fragmented bacilli | 101/131 (77.1%) | Summary in Results |
| Histopathology | Skin/nerve biopsy; H&E and Fite-Faraco | Categories: regression, bacillary persistence, active disease | 123/131 (93.9%) | Definitions per Methods; results summarized in Results |
| qPCR for *M. leprae* | DNA from biopsy; duplicate runs | CT < 40 in duplicate = positive | 101/131 (77.1%) | Summary in Results |
| Drug-resistance genotyping (*folP1, rpoB, gyrA*) | Targeted sequencing | Mutations per WHO/consensus catalogues | 88/131 (67.2%) | Counts by locus/coverage in Results |
| Nude mice inoculation | Homogenized lesion suspension: outcome read per protocol | Positive/negative/ inconclusive outcome within the study window | 45/131 (34.4%) | Concordance with histopathology in Table 5 |
| Disability grade (SNA) | Clinical exam | WHO grade 0/1/2 | 117/131 (89.3%) | Summary in Results |

Denominators reflect only cases with valid results (available-case analysis).

subsections below. For the overlap illustration, each domain was coded as present/absent at the post-MDT index visit: persistent skin lesions, neurological deterioration (new/worsening peripheral symptoms or signs), and reactional episodes with neuritis (at discharge or within 6 months). The Venn diagram (Fig 1) was constructed from patient-level data in the n = 131 clinical cohort; exact counts are listed in Table 4.

### Definition of MDT-therapeutic failure

Following Brazilian MoH guidance, we considered MDT therapeutic failure in MB patients after completion of 24 MDT doses when there was clinical evidence of ongoing activity (operationalized as erythematous/infiltrated lesions, persistent/recurrent reactions, neurological complaints, or worsening on the Simplified Neurological Assessment) and/or histopathological evidence of viable bacilli in skin or nerve biopsy; when available, bacteriological support (e.g., intact bacilli on slit-skin smear) and serology were considered. These criteria match the national description of post-MDT therapeutic failure and are distinct from insufficient treatment and relapse as defined in the same manual [23].

### Clinical sampling at the post-MDT index re-evaluation

The study was conducted at SEINPE (Petrolina Infectious Diseases Service, Petrolina, Pernambuco, Brazil), where patients with suspected post-MDT persistent disease after completing 24-dose MDT are routinely investigated. At the index re-evaluation, the protocol comprised skin biopsy of erythematous or infiltrated lesions, when present; otherwise, biopsy was obtained from skin over thickened peripheral nerves that were tender to palpation or from areas showing abnormal sensation on Semmes–Weinstein monofilament testing.

### Neurological Assessment (SNA) and Disability Grade (WHO 0–2)

Sensory testing employed Semmes–Weinstein monofilaments, and motor/reflex assessments were performed as per the Simplified Neurological Assessment (SNA). The outcome was the WHO disability grade (0, 1, or 2), hereafter referred to as the WHO disability grade, derived from SNA findings at diagnosis and at the post-MDT index visit.

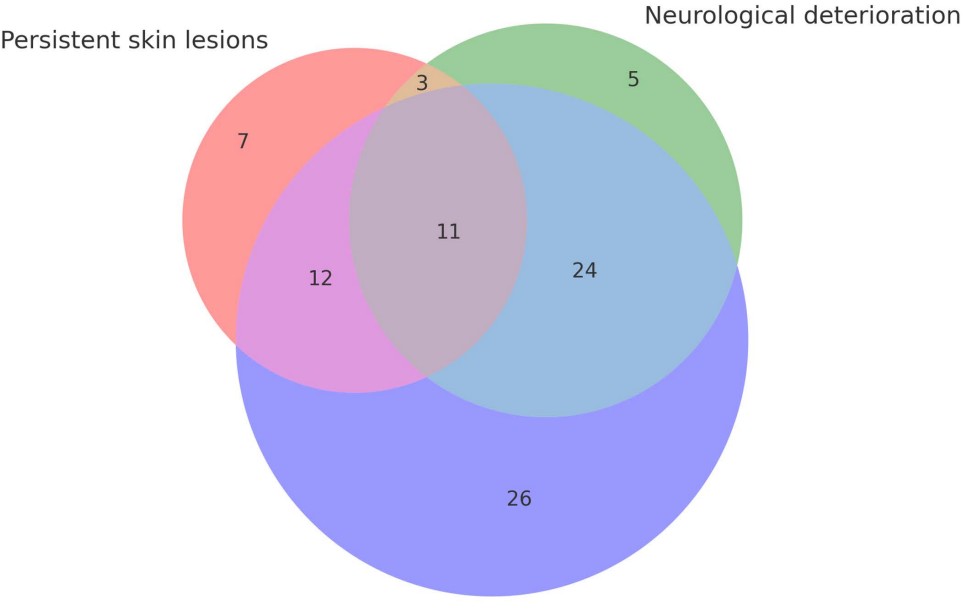

Overlap of clinical domains among patients with paired SNA (n=99)

**Fig 1. Overlap of clinical domains among patients with paired Simplified Neurological Assessment (SNA) at the post-MDT index visit (n=99).** The diagram shows the distribution of patients fulfilling three domains: persistent skin lesions (33/99, 33.3%), neurological deterioration defined as any increase in WHO disability grade from diagnosis to the index visit (43/99, 43.4%), and leprosy reaction with neuritis occurring at or within six months of discharge (73/99, 73.7%). A total of 11/99 (11.1%) patients met all three domains simultaneously; 24/99 (24.2%) presented deterioration with neuritis without skin lesions; 12/99 (12.1%) had skin lesions with neuritis without deterioration; 3/99 (3.0%) had skin lesions with deterioration without neuritis; and 11/99 (11.1%) had none of the three domains.

## Overlap definitions

For the overlap analysis, each domain was coded as present/absent at the post-MDT index visit among patients with paired SNA (n=99): persistent skin lesions; neurological deterioration (any increase in WHO disability grade [0–2] from diagnosis to index); and reaction with neuritis (type 1/2 reaction at discharge or ≤6 months).

## Collection of biological material

At the Leprosy Reference Center, where this study was conducted, it is a standard medical protocol to collect samples for histopathological examination and investigation of drug resistance after completing 24 MDT doses in case of suspected active disease.

Skin biopsies were taken immediately after discharge from 24 regular doses of MDT or within a maximum of six months after treatment completion. Fragments were collected from erythematous/infiltrated lesions when present. In the absence of skin lesions, samples were taken from clinically normal skin overlying a thickened or painful nerve, following the Brazilian Medical Association's primary neural leprosy guidelines [23].

The nerves were selected by identifying neural thickening on medical palpation in the anatomical site of the peripheral nerves commonly affected by leprosy, prioritizing the ulnar, fibular, or tibial nerves.

The surgical technique employed consisted of selecting and marking the biopsy site with a surgical pen, performing asepsis and antisepsis of the surgical field, applying sterile drapes, and infiltrating the area with 2% lidocaine with epinephrine. The skin fragments were removed using a 5mm punch. After suturing the skin with 3–0 nylon, the area was cleaned, and an occlusive sterile dressing was applied.

## Slit skin smear

In Brazil, the standard protocol recommended collecting slit-skin smears from predefined anatomical sites. Therefore, samples were obtained from the right earlobe, left earlobe, right elbow, and from a representative skin lesion. In cases where no evident lesion was present, the left elbow was also sampled to maintain protocol standardization. Smears were stained using the Ziehl–Neelsen technique and examined under light microscopy. The bacilloscopic index (BI) was determined according to Ridley's logarithmic scale, which classifies bacillary load from 0 to 6+: 0 indicating absence of bacilli in 100 fields; 1+ presence of 1–10 bacilli in 100 fields; 2+ presence of 1–10 bacilli per 10 fields; 3+ presence of 1–10 bacilli per field; 4+ presence of 10–100 bacilli per field; 5+ presence of 100–1,000 bacilli per field; and 6+ more than 1,000 bacilli per field, on average [24].

## Histopathological examination

Formalin-fixed tissues were embedded in paraffin, processed for hematoxylin-eosin and Fite-Faraco staining, and examined by an experienced pathologist.

Histopathological findings were classified as follows: disease regression, defined by the presence of fragmented bacilli with variable staining intensity by the Fite-Faraco technique in different tissues or parasitized cells, except within endothelial cells; persistent bacilli, characterized by the presence of well-stained, solid-appearing bacilli within any tissue or parasitized cells after treatment completion; active disease, defined by the presence of intact bacilli in tissues and parasitized cells and/or the presence of intact or fragmented bacilli within endothelial cells after treatment completion. A dense lymphoplasmacytic and histiocytic inflammatory infiltrate involving vessels and nerve branches was considered a histological indicator of disease reactivation due to bacillary proliferation. Additionally, a nonspecific lymphocytic infiltrate, distributed perivascularly, perifollicularly, and/or perineurally, was observed, along with normal skin without histopathological alterations.

## Nude mice inoculation

The biopsies from patients (skin lesions) were homogenized in Hank´s balanced salt solution. The bacilli were enumerated on a glass slide after Ziehl–Neelsen (ZN) staining. The obtained bacillary suspension was injected intradermally into the hind footpads of athymic mice. Animals were maintained for 150 days in the Animal facility. The animal was humanely euthanized following institutional and ethical guidelines, ensuring minimal or no suffering, footpads collected, cut into smaller pieces using scissors, transferred to tubes containing 1000 µL of saline solution, and homogenized by three pulses of a tissue homogenizer (Tissue Homogenizer Omni TH) at a speed of 4 (14,450 rpm) for 15 s. The tubes were always maintained on ice. Then, the homogenates were filtered through a cell strainer to eliminate the remaining debris. Bacilli were enumerated in glass slides after ZN staining.

## qPCR analysis

qPCR was performed in duplicates using 10 ng of total DNA extracted from skin biopsies, using SYBR Green chemistry and the pair of primers (sense 5′ ATTTCTGCCGCTGGTATCGGT 3′, antisense 5′ TGCGCTAGAAGGTTGCCGTAT 3′) (ThermoFisher Scientific, Waltham, MA, USA) as described by Azevedo et al. [25]. Samples were considered positive when they showed a $C_T$ value lower than 40 (cutoff).

## *M leprae* sequencing

Drug resistance acessment followed the protocol stablished in Brazil. The target regions of the *folP1* (gene ID: 908646), *rpoB* (gene ID: 910599), and *gyrA* (gene ID: 908154) genes are available under GenBank accession no. NC002677, and was used as a standard. The sequences were aligned using MEGA7, Molecular Evolutionary Genetics Analysis version 7.0.

## Bias and data completeness

Referral-center spectrum/selection bias was addressed by reporting the cohort as a census of all eligible patients in the period, using uniform operational definitions and displaying denominators for each analytic subset. Historical records at diagnosis/

discharge were variably available, particularly for patients treated outside the referral center; therefore, paired completeness was not required to avoid non-random complete-case selection. Resistance testing and nude mice inoculation were available for subsets due to laboratory capacity and archival tissue constraints; these limitations were reported with their denominators.

### Study size

Because the study comprised all eligible patients during the defined period (census), no formal sample size calculation was performed. Precision was instead addressed by reporting exact confidence intervals for key proportions, and no between-group comparisons were prespecified.

### Handling of missing data

An available-case approach was used with explicit denominators for each estimate; no imputation was performed.

### Statistical analysis

All data were coded and organized in spreadsheets and analyzed using JASP (version 0.17.1.0; JASP Team, University of Amsterdam, The Netherlands). Analyses were descriptive. Categorical variables were summarized as counts and percentages with exact binomial 95% confidence intervals. Continuous measures, when applicable, were summarized as median (IQR). Any post hoc subgroup contrasts (e.g., by resistance status) were labeled exploratory and were reported as effect-size summaries (absolute differences with 95% CIs; standardized differences) without p-values, to be interpreted as hypothesis-generating only.

## Results

Proportions and distributions for each variable are reported below, using the analytic denominators summarized in Table 1. Persistent skin lesions were defined as dermatological lesions present at discharge; neurological deterioration was defined as either worsening of WHO disability grade measurements or neurological symptoms at discharge; and reactional episodes with neuritis were identified according to clinical records.

### Study population and sample availability

A total of 131 multibacillary leprosy patients met the inclusion criteria. Eight biopsy specimens were excluded for technical conservation issues, leaving 123 samples for histopathology. At the post-MDT index evaluation, slit-skin smear for BI was available in 126 cases and the morphological index in 74. Molecular testing comprised qPCR in 101 patients and drug-resistance sequencing in 88 patients (both in 88 patients). Nude mice inoculation yielded an ascertainable outcome in 45 cases.

### Sociodemographic and clinical characteristics

Of 131 patients, 98 (74.8%) were male. The mean age was 45.3 ± 14.2 years (range, 8–84), with a peak in the 40–59-year age bracket. The interval from diagnosis to biopsy averaged 2.3 ± 1.1 months (range, 0–6 months). By Ridley–Jopling [26], most cases were borderline or borderline–lepromatous. Primary neuritic leprosy was defined according to Pannikar et al [27]. The mean BI declined from 2.6 ± 1.3 at diagnosis to 1.5 ± 1.2 after MDT. Detailed socio-demographic and age distributions are shown in Table 2.

Reactional episodes were common; only 6/131 (4.6%) had none, and neuritis was present in 94/131 (71.8%) of the cases. WHO disability grade (assessed via SNA) shifted upward from 1 [0–1] at diagnosis (N = 100) to 1 [1–1] post-MDT (N = 117); means were 0.66 ± 0.57 and 1.09 ± 0.61, respectively (available-case), indicating overall deterioration. Detailed profiles are shown in Table 3.

**Table 2. Socio-demographic and clinical characteristics.**

| Sex | n | % |
|---|---|---|
| Female | 33 | 25.19% |
| Male | 98 | 74.81% |
| Total | 131 | 100% |
| Age stratification | | |
| <10 years old | 1 | 0.76% |
| 10-19 years old | 5 | 3.82% |
| 20-29 years old | 11 | 8.40% |
| 30-39 years old | 12 | 9.16% |
| 40-49 years old | 30 | 22.90% |
| 50-59 years old | 30 | 22.90% |
| 60-69 years old | 26 | 19.85% |
| > 70 years old | 16 | 12.21% |
| Total | 131 | 100% |
| Clinical classification (Ridley-Jopling) | | |
| Borderline | 54 | 41.22% |
| Borderline-lepromatous | 49 | 37.40% |
| Lepromatous | 21 | 16.03% |
| Primary Neuritic Leprosy | 7 | 5.34% |
| Total | 131 | 100% |

## Clinical and dermatological findings

Of 131 patients, 42/131 (32.1%) had persistent erythematous/infiltrated skin lesions at the post-MDT index visit. Peripheral neurological symptoms increased from 70/131 (53.4%) at diagnosis to 114/131 (87.0%) post-MDT. Among patients with paired SNA (n=99), 33/99 (33.3%) had persistent skin lesions, 43/99 (43.4%) showed worsening in their WHO disability grade, and 73/99 (73.7%) experienced a leprosy reaction with neuritis. Fig 1 illustrates the overlap of these domains, with 11/99 (11.1%) patients fulfilling all three simultaneously (see the figure legend for further details). In the whole cohort, leprosy reactions (Type 1/2) occurred in 113/131 (86.3%), and neuritis in 94/131 (71.8%) at or within six months of discharge (Table 3). Despite improvement in skin lesions post-MDT, neurological status generally worsened (WHO disability grade $0.66 \pm 0.57 \rightarrow 1.09 \pm 0.61$; Table 4).

## Laboratory features

**Slit Skin Smear (BI/MI).** At diagnosis, slit-skin smear was not performed in 14/131 patients (10.7%), primarily due to referral from other services; after 24 doses of MDT, 5/131 (3.8%) declined the procedure. Thus, the bacteriological index (BI) was available in 117 patients at diagnosis and in 126 patients post-MDT, while the morphological index (MI) was performed in 70 patients at diagnosis and in 74 patients post-MDT.

**Bacteriological Index (BI).** At diagnosis, 22/117 (18.80%) were BI-negative; post-MDT, the number of negatives increased to 31/126 (24.60%). High bacillary load (BI ≥ 3+) decreased from 60/117 (51.28%) to 25/126 (19.84%). The full BI distribution was as follows: BI 0.25–1, 11/117 (9.40%) → 24/126 (19.05%); BI 1–2, 6/117 (5.13%) → 24/126 (19.05%); BI 2–3, 18/117 (15.38%) → 22/126 (17.46%); BI 3–4, 29/117 (24.79%) → 24/126 (19.05%); BI > 4, 31/117 (26.50%) → 1/126 (0.79%).

**Bacillary Morphology (MI).** Among smears with MI performed, 67/70 (95.71%) at diagnosis showed intact bacilli and/or globi, and 3/70 (4.29%) fragmented/granular forms; post-MDT, 72/74 (97.30%) showed intact/globi and 2/74 (2.70%) fragmented/granular forms.

**Table 3. Reactional profiles at the post-MDT index visit (mutually exclusive).**

| Leprosy reaction classification | n | % |
|---|---|---|
| Leprosy reaction Type 1 | 3 | 2.29% |
| Leprosy reaction Type 2 | 12 | 9.16% |
| Mixed Leprosy reactions type 1 and type 2 | 16 | 12.21% |
| Neuritis | 36 | 27.48% |
| Leprosy reaction Type 1 with neuritis | 15 | 11.45% |
| Leprosy reaction Type 2 with neuritis | 26 | 19.85% |
| Mixed Leprosy reactions type 1 and type 2 with neuritis | 17 | 12.98% |
| No reactions | 6 | 4.58% |
| Total | 131 | 100% |

Categories are mutually exclusive and derived from subtype flags (Type 1, Type 2) and neuritis at discharge or ≤6 months post-MDT. Neuritis indicates clinical neuritis requiring treatment; isolated neuritis refers to neuritis without a Type 1/2 reaction. No reactions=absence of Type 1, Type 2, and neuritis. Available-case approach: denominators are n=131 unless otherwise specified; missing values were not imputed.

## Histopathology

Eight specimens were excluded due to technical conservation issues. Among the remaining 123 index biopsies, regressive disease was observed in 43/123 (34.96%, 95% CI 26.6–44.1), persistent bacilli in 41/123 (33.33%, 95% CI 25.1–42.4), active lesions in 21/123 (17.07%, 95% CI 10.9–24.9), nonspecific perivascular/perifollicular/perineural inflammatory infiltrates in 12/123 (9.76%, 95% CI 5.1–16.4), and normal skin in 6/123 (4.88%, 95% CI 1.8–10.3). Evidence of active disease (defined as either active lesion or persistent, well-stained bacilli) was present in 62/123 (50.41%, 95% CI 41.2–59.5).

Representative histopathological findings consistent with post-treatment persistence are shown in Fig 2.

At diagnosis, slit-skin smear was not performed in 14 patients, whereas histopathology could not be assessed in 8 specimens due to technical failure. These groups did not overlap, meaning that patients lacking a slit-skin smear at diagnosis were not the same individuals as those with missing histopathology. Additionally, a few patients also lacked a slit-skin smear at the time of discharge, as reported separately in the follow-up results. Despite these baseline and follow-up limitations, all 131 patients were retained in the analysis, as their clinical and laboratory data remained essential for describing post-treatment outcomes.

## Molecular detection and drug resistance

qPCR was performed in 101/131 (77.10%) samples; 96/101 (95.05%; 95% CI 88.9–97.9) were positive for *M. leprae* DNA, and 5/101 (4.95%; 95% CI 2.1–11.1) were negative. The remaining 30/131 (22.90%) were not tested, primarily due to insufficient material, logistics constraints, or patient refusal.

Drug-resistance genotyping of *folP1*, *rpoB*, and *gyrA* succeeded in 88/131 (67.18%) patients, yielding 85/88 (96.59%; 95% CI 90.5–98.8) susceptible, 2/88 (2.27%; 95% CI 0.6–7.9) resistant, and 1/88 (1.14%; 95% CI 0.2–6.2) inconclusive. Resistance-associated mutations were detected in *gyrA* (1/88; 1.14%; 95% CI 0.2–6.2) and *folP1* (1/88; 1.14%; 95% CI 0.2–6.2); no *rpoB* mutations were found (0/88; 0.00%; 95% CI 0.0–4.2). The remaining 43/131 (32.82%) were not genotyped for the same operational reasons.

Interestingly, mutations not yet validated as resistance determinants were detected in 16 of 88 (18.18%) samples. Because these variants have not yet been functionally linked to drug resistance in *M. leprae*, we report them descriptively (Table 4) without inferring resistance or discussing clinical implications.

**Table 4. Clinical and laboratory characteristics at diagnosis and after completion of 24-dose MDT (post-MDT index re-evaluation).**

**Clinical Features**

| Dermatological lesions, symptoms, and leprosy reactions | At the time of diagnosis n/N (%) | Following 24 doses of MDT n/N (%) |
|---|---|---|
| Dermatological lesions —present | 124/131 (94.66%) | 42/131 (32.06%) |
| Dermatological lesions — absent | 7/131 (5.34%) | 89/131 (67.94%) |
| Neurological symptoms —present | 70/131 (53.44%) | 114/131 (87.02%) |
| Neurological symptoms —absent | 61/131 (46.56%) | 17/131 (12.98%) |
| Leprosy reactions (Type 1/2) present | 120/131 (91.60%) | 113/131 (86.26%) |
| Leprosy reactions (Type 1/2) absent | 11/131 (8.40%) | 18/131 (13.74%) |

**Simplified neurological assessment**

| WHO disability grade measurement | At the time of diagnosis, n/N (%) (n = 100) | Following 24 doses of MDT, n/N (%) (n = 117) |
|---|---|---|
| WHO disability grade 0 | 39/100 (39.00%) | 17/117 (14.53%) |
| WHO disability grade 1 | 56/100 (56.00%) | 73/117 (62.39%) |
| WHO disability grade 2 | 5/100 (5.00%) | 27/117 (23.08%) |
| WHO disability not performed | 31/131 (23.66%) | 14/131 (10.69%) |

**Laboratory features**

| Slit skin smear and bacteriological index (BI) | At the time of diagnosis n/N (%) (n = 117) | Following 24 doses of MDT n/N (%) (n = 126) |
|---|---|---|
| Negative | 22/117 (18.80%) | 31/126 (24.60%) |
| BI 0.25 – 1 | 11/117 (9.40%) | 24/126 (19.05%) |
| BI 1 – 2 | 6/117 (5.13%) | 24/126 (19.05%) |
| BI 2 – 3 | 18/117 (15.38%) | 22/126 (17.46%) |
| BI 3 – 4 | 29/117 (24.79%) | 24/126 (19.05%) |
| BI > 4 | 31/117 (26.50%) | 1/126 (0.79%) |
| Slit skin smear not performed | 14/131 (10.69%) | 5/131 (3.82%) |

| Slit skin smear and morphological aspect | At the time of diagnosis n/N (%) (n = 70) | Following 24 doses of MDT, n/N (%) (n = 74) |
|---|---|---|
| Intact bacilli and (or) globi (present) | 67/70 (95.71%) | 72/74 (97.30%) |
| Intact bacilli and (or) globi (absent) | 3/70 (4.29%) | 2/74 (2.70%) |
| Fragmented or granular bacilli (present) | 3/70 (4.29%) | 2/74 (2.70%) |
| Fragmented or granular bacilli (absent) | 67/70 (95.71%) | 72/74 (97.30%) |
| Morphological aspect not performed | 47/117 (40.17%) | 52/126 (41.27%) |

| Histopathology | Following 24 doses of MDT n/N (%) (n = 123) | |
|---|---|---|
| Normal skin | 6/123 (4.88%) | |
| Regressive disease | 43/123 (34.96%) | |
| Persistent bacilli | 41/123 (33.33%) | |
| Active lesion | 21/123 (17.07%) | |
| Nonspecific perivascular and (or) perifollicular and (or) perineural inflammatory infiltrate | 12/123 (9.76%) | |
| Histopathology not performed | 8/131 (6.11%) | |

*(Continued)*

**Table 4.** (Continued)

| Clinical Features | | |
|---|---|---|
| Dermatological lesions, symptoms, and leprosy reactions | At the time of diagnosis n/N (%) | Following 24 doses of MDT n/N (%) |
| qPCR result | Following 24 doses of MDT n/N (%) (n = 101) | |
| Positive<br>Negative<br>qPCR not performed | 96/101 (95.05%)<br>5/101 (4.95%)<br>30/131 (22.90%) | |
| Drug Resistance Test and Gene Mutations | Following 24 doses of MDT n/N (%) (n = 88) | |
| A) Overall test results<br>Susceptible<br>Resistant<br>Inconclusive<br>Not performed<br>B) Gene mutations<br>*rpoB* (rifampicin resistance)<br>*gyrA* (ofloxacin resistance)<br>*folP1* (dapsone resistance)<br>Other mutations (not yet validated for drug resistance)* | 85/88 (96.59%)<br>2/88 (2.27%)<br>1/88 (1.14% )<br>43/131 (32.82%)<br>0/88 (0.0%)<br>1/88 (1.1%)<br>1/88 (1.1%)<br>16/88 (18.18%) | |
| Nude mice inoculation test | Following 24 doses of MDT n/N (%) (n = 45) | |
| Positive<br>Negative<br>Inconclusive<br>Nude mice inoculation test | 29/45 (64.44%)<br>10/45 (22.22%)<br>6/45 (13.33%)<br>86/131 (65.65%) | |

a. Available-case analysis: denominators are cell-specific (only observed values); no imputation. "At diagnosis" = baseline; "Following 24 doses of MDT" = post-MDT index visit.
b. Denominators by block: BI 117/126; MI 70/74; Histopathology 123; qPCR 101; Drug-resistance testing 88; nude mice inoculation 45. Rows labeled "not performed" generally use the full cohort (n = 131), except for MI "not performed," where denominators correspond to patients with BI performed (diagnosis n = 117; post-MDT n = 126).
c. Leprosy reactions (Type 1/2) refer only to reactional episodes; neuritis is not included here (the phenotypic breakdown with neuritis is shown in Table 2).
d. WHO disability grade was assessed via the SNA; grades 0–2.
e. Assay definitions: BI categories per Methods (the "0.25–1" class merges formatting variants); MI reported only among patients with MI performed (intact bacilli and/or globi vs. fragmented/granular); qPCR positive when Ct < 40 in duplicate; drug-resistance genotyping targeted *folP1*, *rpoB*, and *gyrA* ("other mutations" = non-validated variants); nude mice inoculation results were counted only when ascertainable (positive/negative/inconclusive).
* These patients were considered susceptible.

## Nude mice inoculation

Inoculation in nude mice was performed in 45 patients due to limited animal availability and operational constraints related to the timely receipt and processing of samples. Positive footpad growth occurred in 29/45 (64.44%; 95% CI 49.8–76.8); 10/45 (22.22%; 95% CI 12.5–36.3) were negative; and 6/45 (13.33%; 95% CI 6.3–26.2) were inconclusive. Not performed: 86/131 (65.65%).

Fig 3 summarizes the clinical and bacteriological evolution after 24 MDT doses, and Table 4 summarizes the clinical and laboratory characteristics at diagnosis and after completion of the 24-dose MDT. Histopathology and nude mice inoculation were conducted only at the post-MDT index evaluation.

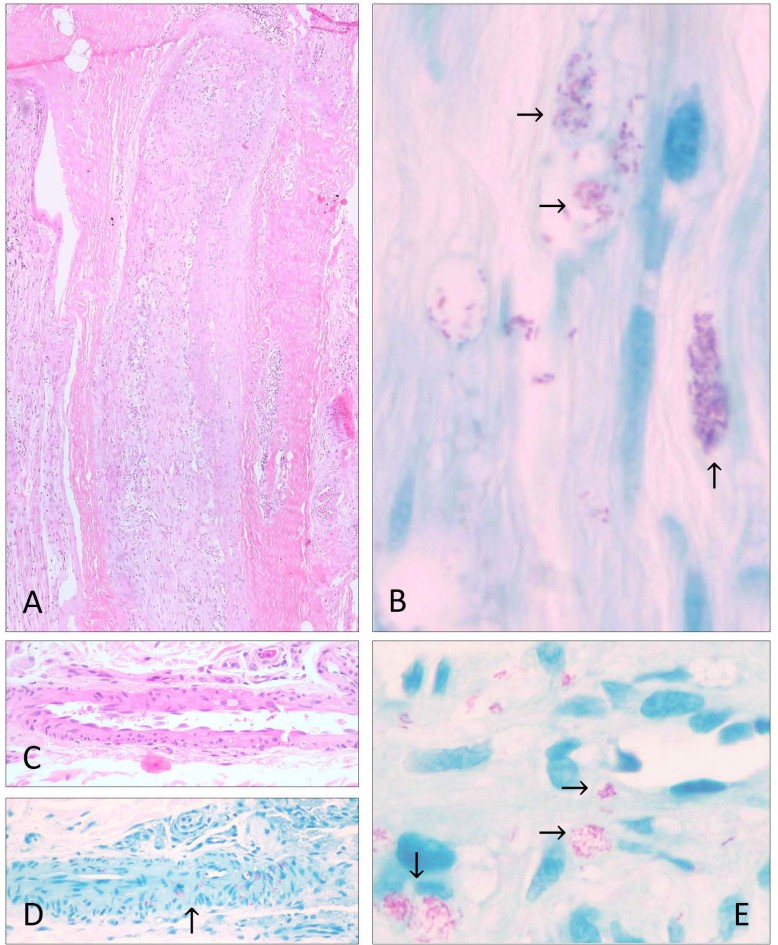

**Fig 2. Case 1 (A and B): After the treatment (24 doses of MDT), neural branches with an intense peri and endoneural lymphohistiocytic inflammatory process were observed (A) (→).** The histopathological bacilloscopic examination showed numerous well-stained fragmented bacilli and rare solid bacilli inside the Schwann cells (B) (→). Case 2 (C, D, and E): after 24 doses of MDT, a skin biopsy shows numerous well-stained fragmented bacilli and some solid bacilli in the vessel walls and endothelium D and E) (→). H&E in A (×20) and C (×20). Fite-Faraco in B (×20), D (×20), E (×40).

**Nude mice inoculation and concordance with histopathology.**

Of the 131 biopsy specimens, 123 underwent histopathology, 45 were inoculated in nude mice, and 101 were tested by qPCR (RLEP). Technical issues precluded histopathology in 8 specimens; 4 of these were nevertheless inoculated. We assessed histology–inoculation concordance in the 41 cases with both assessments.

Overall, positive footpad growth occurred in 29/45 (64.4%; 95% CI 49.8–76.8); 10/45 (22.2%; 95% CI 12.5–36.3) were negative; and 6/45 (13.3%; 95% CI 6.3–26.2) were inconclusive.

Among the 29 positives, histopathology showed persistent, well-stained bacilli in 15 (51.7%), active lesions in 5 (17.2%), and regressive features (granular bacilli/bacillary dust) in 6 (20.7%); histopathology was not available in 3 (10.3%). Of the 29 inoculation-positive cases, histopathology was available in 26; among these, persistent bacilli were found in 15/26 (57.7%), active lesions in 5/26 (19.2%), and regressive features in 6/26 (23.1%). Table 5 contains the full histopathology–inoculation cross-tabulation.

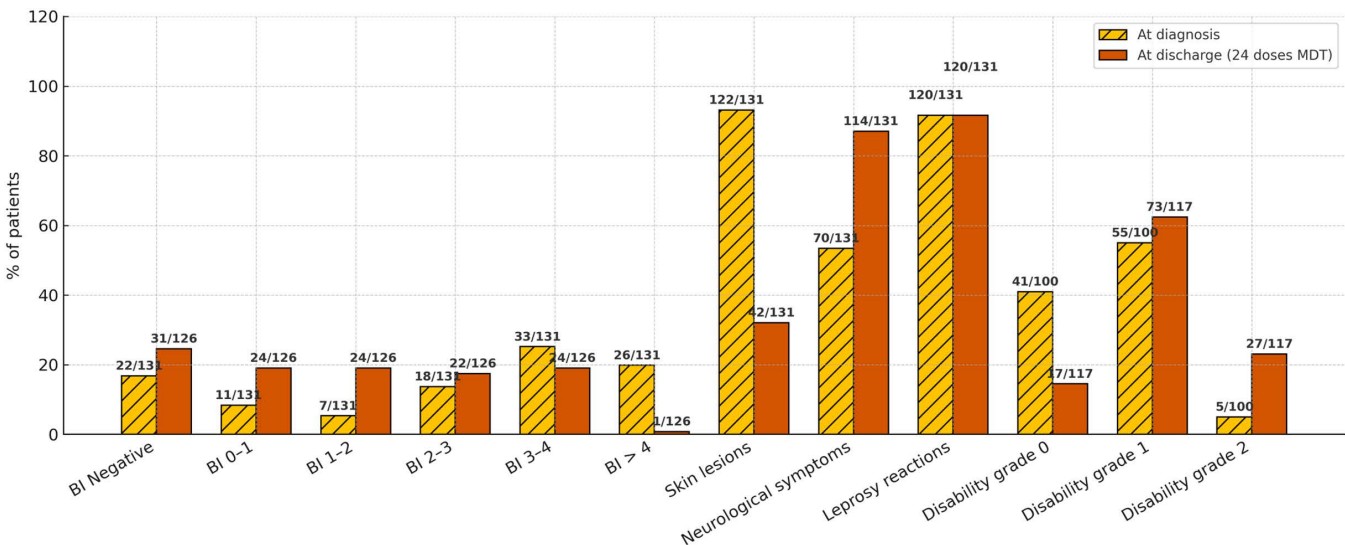

**Fig 3. Proportions of patients at diagnosis and after completing 24 doses of MDT are shown.** Bacilloscopic index (BI) categories follow Ridley's scale; denominators for BI and morphological index correspond only to patients with a valid slit-skin smear at each time point. Skin lesions, neurological symptoms, and leprosy reactions refer to the entire cohort (n = 131). Disability grades follow WHO classification (0/1/2); denominators differ between baseline (n = 100) and post-MDT (n = 117) due to availability of paired SNA. No imputation was performed.

Row totals refer to specimens with available histopathology (n = 41). The "No histopathological examination" row comprises the four specimens inoculated without undergoing histological examination. The table total (n = 45) corresponds to all inoculations with an ascertainable outcome (positive/negative/inconclusive). Percentages are calculated within each row; in the bottom "Total" row, percentages are calculated out of a sample size of n = 45.

## Discussion

The introduction of WHO-endorsed multidrug therapy (MDT) in the early 1980s resulted in a dramatic reduction in global leprosy prevalence. However, new case numbers have plateaued in major endemic countries such as Brazil and India over the past decade, disregarding the post-pandemic period, during which there was a dramatic reduction in the number of diagnoses. In Brazil alone, there was a 41.4% decrease in the total number of new cases, reaching 56.82% among children under 15 years of age. Spatial analysis indicates a 100% reduction in some Brazilian states, with recovery remaining incomplete, with long-term epidemiological impacts still to be defined. [28,29].

In our series of 131 multibacillary patients treated with an extended 24-month MDT regimen, twice the yet recommended 12-month course [30], 52.0% retained histological or microbiological evidence of *M. leprae* activity despite marked clinical improvement. This descriptive finding highlights the limited sensitivity of skin-lesion clearance as a cure marker and raises concerns about relying solely on fixed-duration treatment regimens.

Mechanistically, our data reveal a dissociation between cutaneous and neural outcomes. Although 35.0% of biopsy specimens exhibited regressive histopathological changes (granuloma involution, bacillary fragmentation, or preserved architecture), most patients nonetheless suffered neurological deterioration, with peripheral complaints rising from 54.2% at diagnosis to 87.0% post-MDT and 24.2% showing worsened SNA.

These observations echo Menicucci et al.'s report of non-specific mononuclear infiltrates in clinically normal, hypoesthetic skin [31] and Joshi's description of heterogeneous lesion stages on post-MDT biopsy [32]. Our study extends these

**Table 5.** Concordance between histopathology and nude mice inoculation results.

| Histopathology result | Total n | Positive n (%) | Negative n (%) | Inconclusive n (%) |
|---|---|---|---|---|
| Regressing disease | 12 | 6 (50.0%) | 5 (41.70%) | 1 (8.30%) |
| Persistent bacilli | 22 | 15 (68.20%) | 4 (18.20%) | 3 (13.60%) |
| Active lesion | 7 | 5 (71.4%) | 0 (0.00%) | 2 (28.60%) |
| No histopathological examination | 4 | 3 (75.00%) | 1 (25.00%) | 0 (0.00%) |
| Total (ascertainable inoculation) | 45 | 29 (64.40%) | 10 (22.20%) | 6 (13.30%) |

insights by sampling skin above thickened nerves, confirming that quiescent bacilli within neural tissues can evade dermatological detection despite driving the immunopathology.

Molecular assays and *in vivo* inoculation further corroborate residual infectivity, with a striking 95.05% of specimens testing positive for *M. leprae* DNA by qPCR. However, the technique cannot distinguish dead from viable bacilli. Athymic nude mice inoculation confirmed the presence of live organisms in 64.44% of the analyzed samples. The concordance analysis shows that even samples judged regressive histologically yielded viable bacilli in 50.0% of cases, and persistent-bacilli samples grew in 68.20%. These results parallel Save et al.'s demonstration that viable *M. leprae* persists in Type 1 reaction lesions post-MDT and correlates with inflammatory activity [33]. Taken together, the data argue for a multidimensional monitoring strategy—serial qPCR quantification, targeted histopathology, and inoculation studies—to capture residual bacilli's burden and viability.

Persistent infection patterns also defy simple associations with initial bacterial index (BI). Early studies by Avelleira et al. (2003) and Celona et al. (2003) linked post-MDT bacillary persistence predominantly to patients with BI ≥ 3+ [34,35], and Brito et al. (2008) connected high BI and ML-Flow seropositivity to recurrent Type 2 reactions [36]. Yet, in our cohort, 96.17% of patients had BI-positive results at discharge, and 64.44% of the 45 patients still harbored active bacilli post-treatment, according to nude mice inoculation results. However, at diagnosis, only 23,66% had BI > 4.0. This finding suggests that factors beyond high bacillary load, such as bacillary dormancy, host-pathogen immune interactions, or pharmacokinetic variability, play a critical role in therapeutic failure. De Carvalho Dornelas et al.'s identification of histological BI ≥ 1+ and qPCR positivity as independent predictors of failure, culminating in a 95.4% combined risk, further emphasizes the multifactorial nature of relapses [37].

Clinical management implications are profound. Van Brakel et al. (1996) showed partial neural recovery with three months of corticosteroids followed by a plateau [38], and Balagon et al. (2010) found reduced reaction rates with 24 months of MDT [39]. Our patients, however, despite extended MDT and up to six months of steroid therapy, displayed persistent or worsening neuritis. These data reinforce that immunomodulation alone cannot substitute for the effective eradication of nerve-resident bacilli through antimicrobial treatment.

Furthermore, historical MDT implementation lapses—such as monthly rifampicin dosing of only two capsules—undoubtedly contributed to suboptimal bactericidal pressure and underscore the necessity of daily rifampicin as was practiced by the U.S. National Hansen's Disease Program (NHDP), which mandated 600mg/day rifampicin and extended treatment to maximize efficacy [40].

In our study, over 50% of patients continued to harbor active *M. leprae* despite marked clinical and dermatological improvement following an extended 24-dose MDT regimen. This observation closely parallels the findings of El-Darouti et al. (2006), which verify that the length of treatment did not show a significant influence on the nature of pathological findings in clinically normal-appearing skin of leprosy patients [41].

Moreover, Gelber (2004) had already warned that the relapse rate of leprosy in patients treated with 24 doses of MDT is not low. However, he concluded that relapses would be confined to groups classified as borderline-lepromatous and lepromatous, with high BIs at the time of diagnosis [42]. Our study, however, demonstrated that even patients with a low

bacillary load, or even with a negative slit skin smear, failed to achieve cure of the disease, despite having been treated with 24 doses of MDT.

A key clinical finding of our study was the persistence of leprosy reactions and neuritis after 24 months of MDT. Save et al. (2016) [33] showed that type 1 reactions and neuritis indicate bacillary proliferation and active disease. Although type 2 reactions were not assessed, other reports suggest that retreatment of patients with persistent reactions after "discharge for cure" can suppress these episodes [43–45], again pointing to ongoing disease activity. Gelber et al. (2010) further noted that a longer duration of MDT was associated with fewer type 1 and type 2 reactions [46].

Together, these data underscore the limitations of relying solely on cutaneous clearance as a cure marker and argue for routine incorporation of tissue-based assessments, whether by repeat biopsy, molecular detection, or mice inoculation to identify quiescent bacillary reservoirs. Such a multidimensional approach may enable earlier identification of patients at risk for relapses or neuropathic progression and support more tailored post-MDT surveillance strategies.

Ultimately, current policy frameworks hinder the recognition and management of persistent diseases. WHO guidelines recommend against routine bacteriological confirmation of cure, discharging patients upon regimen completion, regardless of residual infection [6]. In Brazil, the SINAN notification system lacks fields for therapeutic failure or post-discharge persistence, relegating re-entries to "other" categories and delaying relapse classification until five years post-discharge [47,48].

Such bureaucratic constructs perpetuate unchecked transmission and disability. We advocate for policy revision to integrate post-MDT surveillance, incorporating clinical, histological, molecular, and inoculation metrics, thereby enabling the early detection of therapeutic failure, not relapses, targeted retreatment, and prevention of irreversible neural impairment.

Misclassifying active leprosy after completion of standard treatment as mere reactional episodes and treating it with immunosuppressive agents contributes to the progression of irreversible disabilities [49].

Future prospective studies should evaluate the predictive value of serial qPCR quantification, repeat biopsies, and inoculation in nude mice to guide individualized post-treatment management. Additionally, they should assess the efficacy of adjunctive antimicrobial and immunomodulatory interventions in preventing leprosy persistence and neuropathic progression.

## Study limitations

This case series was restricted to 131 multibacillary patients who, despite completing 24 MDT doses, continued to exhibit refractory leprosy reactions or neurological deterioration and therefore underwent skin biopsy to assess disease activity. As a result of this selection bias, our findings cannot be extrapolated to all treated patients, particularly those who achieved clinical stability and were managed without biopsy or the other tests performed in this study.

The retrospective design further limited our study, as not all patients were followed entirely at the SEINPE referral center, and some did not complete every clinical or laboratory evaluation, resulting in variable denominators across assays and potential underrepresentation of specific findings.

Anti-PGL-1 serology was not systematically collected in this cohort. Although anti-PGL-1 IgM correlates with bacillary load and may assist risk stratification, it does not discriminate bacillary viability, and its performance varies across assays and cut-offs. Given that our framework already integrates histopathology, qPCR, and nude mice inoculation, the incremental value of serology for confirming post-treatment persistence would likely be limited. Nonetheless, future studies should assess longitudinal serology using standardized methods to refine risk profiling.

Data were collected solely from the municipality of Petrolina, a hyperendemic region in Pernambuco, which may limit the generalizability of our results to other geographic or epidemiological contexts.

Although household contacts were evaluated according to routine protocol, we did not explicitly analyze the risk of reinfection in this report. In a setting of sustained transmission, community reinfection cannot be entirely excluded as

a contributor to persistent or recurrent disease activity; however, the magnitude and consistency of our histopathological and microbiological findings suggest that reinfection alone is unlikely to account for the observed treatment failures.

## Conclusion

Our findings demonstrate that a significant subset of multibacillary patients retains active *M. leprae* infection and continues to experience clinical or subclinical disease activity even after completing the recommended MDT course. This underscores the inadequacy of time-based discharge criteria and the absence of standardized post-MDT surveillance protocols at both national and international levels.

Health authorities in Brazil and other endemic countries urgently need to develop evidence-based guidelines for extending follow-up and diagnostic evaluation, including clinical, histopathological, molecular, and *in vivo* assays, in patients who present persistent or recurrent signs of leprosy activity.

By instituting such protocols, we can identify therapeutic failures earlier, tailor retreatment strategies, and thereby reduce the risk of irreversible neuropathy, disability, and associated psychosocial harms. Ultimately, a more rigorous, multidimensional approach to post-treatment monitoring will help preserve patient quality of life, diminish stigma, and curb ongoing transmission.

Drug resistance to MDT was not a significant concern in this study, with only 2.27% of patients showing resistance in routinely tested molecular targets and 1.14% yielding inconclusive results. More concerning was the detection of several previously unreported mutations (18.18%), not yet associated with resistance, which indicates the circulation of novel strains in the community and highlights the need for further investigation.

## Supporting information

**S1 STROBE Checklist. The filled checklist is based on the STROBE Statement-Checklist of items that should be included in reports of observational studies, developed by the STROBE Initiative, https://www.strobe-statement.org/.**
(DOCX)

**S1 CARE Checklist. Adapted from the CARE Statement (https://www.care-statement.org), licensed under CC BY 4.0 (https://creativecommons.org/licenses/by/4.0/).**
(DOCX)

**S1 File. Striking image legend.**
(DOCX)

**S2 File. Striking image - new.**
(TIF)

## Acknowledgments

To the entire team at Instituto Lauro de Souza Lima in Bauru, SP, who directly or indirectly contributed to the completion of this work, and to the Municipal Health Department of Petrolina, PE, for their dedication and effort in providing additional care to patients living with leprosy.

## Author contributions

**Conceptualization:** Andrea Maia Fernandes de Araújo Fonseca.

**Data curation:** Andrea Maia Fernandes de Araújo Fonseca.

**Formal analysis:** Rodrigo Feliciano do Carmo.

**Investigation:** Patrícia Sammarco Rosa, Andrea de Faria Fernandes Belone, Cleverson Teixeira Soares, Daniele de Faria Ferreira Bertoluci, Suzana Madeira Diório, Luciana Raquel Vincenzi Fachin.

**Methodology:** Patrícia Sammarco Rosa, Andrea de Faria Fernandes Belone, Cleverson Teixeira Soares, Daniele de Faria Ferreira Bertoluci, Suzana Madeira Diório, Luciana Raquel Vincenzi Fachin.

**Project administration:** Rodrigo Feliciano do Carmo.

**Resources:** Patrícia Sammarco Rosa, Andrea de Faria Fernandes Belone, Rodrigo Feliciano do Carmo.

**Software:** Patrícia Sammarco Rosa, Andrea de Faria Fernandes Belone.

**Supervision:** Rodrigo Feliciano do Carmo.

**Validation:** Patrícia Sammarco Rosa, Andrea de Faria Fernandes Belone, Cleverson Teixeira Soares, Daniele de Faria Ferreira Bertoluci.

**Visualization:** Cleverson Teixeira Soares.

**Writing – original draft:** Francisco Bezerra de Almeida Neto.

**Writing – review & editing:** Francisco Bezerra de Almeida Neto.

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
