## [Decision Letter · Decision Letter 0]

31 Jan 2025

PNTD-D-24-01924

Multidrug Therapy for Leprosy Fails in an Important City in a Hyperendemic Region of Brazil

Dear Dr. Fonseca,

Thank you for submitting your manuscript to PLOS Neglected Tropical Diseases. After careful consideration, we feel that it has merit but does not fully meet PLOS Neglected Tropical Diseases's publication criteria as it currently stands. Therefore, we invite you to submit a revised version of the manuscript that addresses the points raised during the review process.

Please submit your revised manuscript within 60 days Apr 01 2025 11:59PM. If you will need more time than this to complete your revisions, please reply to this message or contact the journal office at plosntds@plos.org. Please include the following items when submitting your revised manuscript:

We look forward to receiving your revised manuscript.

Kind regards,

Feng Xue, Ph.D.

Guest Editor

Qu Cheng

Section Editor

Shaden Kamhawi

co-Editor-in-Chief

Paul Brindley

co-Editor-in-Chief

**Journal Requirements:**

Please ensure that the CRediT author contributions listed for every co-author are completed accurately and in full.

At this stage, the following Authors/Authors require contributions: Andrea Fonseca, Patrícia Sammarco Rosa, Andrea de Faria Fernandes Belone, Cleverson Teixeira Soares, Daniele Bertolucci, Rodrigo Feliciano do Carmo, and Francisco Bezerra de Almeida Neto. Please ensure that the full contributions of each author are acknowledged in the "Add/Edit/Remove Authors" section of our submission form.

**Reviewers' Comments:**

Reviewer's Responses to Questions

**Key Review Criteria Required for Acceptance?**

**Methods**

-Are the objectives of the study clearly articulated with a clear testable hypothesis stated?

-Is the study design appropriate to address the stated objectives?

-Is the population clearly described and appropriate for the hypothesis being tested?

-Is the sample size sufficient to ensure adequate power to address the hypothesis being tested?

-Were correct statistical analysis used to support conclusions?

-Are there concerns about ethical or regulatory requirements being met?

Reviewer #1: In the introduction, line 27, I suggest not restricting disability to untreated cases.

In the methods, line 163-166, what are the neurological complaints? To conclude that there was a persistence of the neurological complaint or worsening of the neural damage, was the simplified neurological assessment after the 24 doses compared with the assessment performed at diagnosis? The results demonstrate this comparison (diagnosis and after 24 doses), but it must too be described in the methods.

The worsening of neural damage was considered only when there was an increase in the degree of physical disability? It would be interesting analyzing the eye-hand-foot score (sum of disabilities in different segments),as well as comparing the number of nerves affected at diagnosis and after 24 doses.

The idea that the selected patients had persistent leprosy reactions needs to be better defined in the methods. What criteria were used to define the reaction episodes as persistent leprosy reactions? The fact of being in reaction in the clinical evaluation after 24 doses of the multidrug? A specific number of reaction episodes, or recurrent reaction episodes without remission with conventional treatment?

It is necessary to clarify whether the collection of histopathological examinations after 24 doses of treatment is part of the medical protocol of the leprosy reference unit where the study was conducted.

Line 175-176 It is also not usual to collect histopathological examinations in areas without apparent skin lesions; is it part of the routine in patients with 24 doses of treatment? And the investigation of resistance and/or disease activity are also part of the protocol of the health unit in question, given that the study information was obtained by analyzing the patients' medical records. These data need to be further elucidated in the methods

Line 182 which categorical variables were used

Reviewer #2: Are the objectives of the study clearly articulated with a clear testable hypothesis stated?

Yes, the study's objective is clearly articulated: to verify disease activity in patients with persistent symptoms after 24 doses of multidrug therapy (MDT) for leprosy. The hypothesis that MDT may fail in some cases is implied through the investigation of active disease and therapeutic failure.

Is the study design appropriate to address the stated objectives?

Yes, the retrospective, observational, descriptive, cross-sectional design is appropriate for analyzing existing medical records, histopathological findings, and other laboratory tests to determine disease activity after MDT.

Is the population clearly described and appropriate for the hypothesis being tested?

Yes, the study population (131 patients with multibacillary leprosy treated at a Brazilian referral center) is clearly described, including demographic and clinical characteristics. This population is appropriate to test the hypothesis of MDT failure in a hyperendemic region.

Is the sample size sufficient to ensure adequate power to address the hypothesis being tested?

The sample size of 131 patients is reasonably robust for a study of this nature, providing sufficient data to draw meaningful conclusions about MDT's effectiveness in this specific setting.

Were correct statistical analyses used to support conclusions?

Yes, descriptive statistics were used appropriately to present demographic, clinical, and laboratory findings. The use of histopathology and molecular biology methods adds rigor to the study's conclusions.

Are there concerns about ethical or regulatory requirements being met?

No significant concerns were identified. The study adheres to Brazilian ethical guidelines (Resolutions 466/2012 and 510/2016) and was approved by an ethics committee. Patient confidentiality was maintained, and data were handled appropriately.

Reviewer #3: No.

No.

No.

not calculated.

it was not performed.

yes

Reviewer #4: The method requires extensive detail. Here are the main points:

1. Regarding the study design, it is a retrospective case series, not a cross-sectional study as previously mentioned. Therefore, it is unnecessary to state that the study is 'observational.'

2. The authors should include the protocol used for the skin biopsy in the main text or Supplementary Material.

3. In the 'Data Collection and Statistical Analysis' section, please specify all data collected and their sources. For example, clinical data were obtained from notification forms or medical records, and biological samples were collected from which source? Additionally, specify how the quantitative variables were analyzed and whether there was a subgroup analysis.

4. Describe the histopathology protocol in the text or Supplementary Material.

5. How was the qPCR analysis performed? Specify how the samples were collected, stored, and analyzed, including the location of the analysis.

6. How was the analysis of genetic mutations carried out, and which ones were investigated?

7. Specify the drug resistance test: What test was performed, and how was it conducted? This is confusing for the reader, as it does not clarify which test was used.

8. How were the mice inoculated?

9. In the ethical considerations section, the authors did not address ethical issues related to animal experiments.

**Results**

-Does the analysis presented match the analysis plan?

-Are the results clearly and completely presented?

-Are the figures (Tables, Images) of sufficient quality for clarity?

Reviewer #1: The results are also well described and the tables are easy to understand. However, it would be interesting separate type 1 reactions with associated neuritis from type 2 reactions with associated neuritis to have a clearer view of how many had type 1 reactions (with or without neuritis) and how many had type 2 reactions (with or without neuritis).

In table 1 in occurrence of reaction, Type 1 would refer to reactions with only exacerbation of inflammation in skin lesions, without neural damage? and Type 2 with erythema nodosum but without neural damage? And Neuritis would refer in this case to isolated neuritis? And maybe it would be better to put instead of blank, without reactions.

In table 2, It was not clear why only 44 inoculation in nude mice were performed. Operational difficulties?

Reviewer #2: Does the analysis presented match the analysis plan?

Yes, the analysis presented aligns with the study's methodology. The study aimed to assess disease activity using histopathological examination, inoculation in mice, and molecular biology tests. These analyses were performed as planned and provided insights into MDT's effectiveness.

Are the results clearly and completely presented?

Yes, the results are presented in a clear and detailed manner, summarizing key findings:

52% of histopathological examinations showed active disease or bacillary persistence.

Bacillary multiplication was observed in 63.63% of inoculations in nude mice.

67.02% of patients had no dermatological lesions at the end of MDT, yet neurological complaints persisted or worsened in 87% of cases.

Drug resistance was rare (two cases), but 28.57% of patients had genetic mutations not associated with known resistance.

Physical disability levels worsened for a significant proportion of patients despite treatment.

Are the figures (Tables, Images) of sufficient quality for clarity?

Yes, the tables provided are of sufficient quality and clarity to support the findings. They effectively summarize the socio-demographic characteristics, clinical and laboratory findings, and results of various tests conducted during the study. For example:

Table 1 outlines socio-demographic and clinical characteristics.

Table 2 lists the tests conducted and their frequencies.

Table 3 compares clinical and laboratory characteristics at diagnosis and after MDT.

Table 4 correlates histopathology results with inoculation findings.

Suggestions for Improvement:

Adding graphical representations (e.g., bar charts or pie charts) for some key findings could enhance visual clarity.

Including photographs or illustrations of histopathological slides or test procedures might add more depth to the analysis.

Reviewer #3: yes

no

no

Reviewer #4: 1. The sum of the relative frequencies of the categories does not total 100% in most tables and text descriptions. Review and correct the entire results section.

2. What classification of clinical forms of leprosy was used in this study? The authors mention the borderline and borderline lepromatous forms, implying that the classification is based on Ridley & Jopling. However, within this same category, the authors include pure neuritic leprosy. I suggest specifying the classification.

3. In Table 1, replace the term “Blank” with “not available” or “not registered.”

4. The content in lines 225-227 must refer to Table 2.

5. In line 228, remove the term “According to WHO.”

6. In line 236, the authors refer to the term “active disease.” I suggest defining this concept. Additionally, include in the methods a brief explanation of the categories included: regressive disease, persistent bacilli, active lesion, and nonspecific perivascular and/or perifollicular and/or perineural inflammatory infiltrate.

7. In line 240, which other mutations evaluated were not associated with drug resistance? Specify the method.

8. “The most frequent type of leprosy reaction was painful and silent neuritis, followed by erythema nodosum leprosum (ENL)” (224-225) is out of context. I suggest placing this statement next to the paragraph where the authors discuss leprosy reactions in this same section.

9. Line 227: Dermal scraping results differ in total number of participants. There are 71 patients analyzed at diagnosis and 87 patients after 24 doses. Aren’t these the same individuals? Why was the initial n 71? Also, specify those who were not analyzed and the reason. The same issue occurs with other results in Table 3.

10. The authors must present the mean/median, standard deviation, minimum, and maximum for quantitative variables. Categorical variables should include absolute and relative frequencies. For example, Table 3 does not present relative frequencies.

11. Lines 228-233: Include clinical characteristics and results of the simplified neurological assessment along with the clinical characteristics presented in Table 1.

12. In Table 3, place the Yes/No categories in columns for the Dermatological lesions variable.

13. In the introduction, the authors refer to the need to evaluate the disease in individuals whose clinical manifestations persist after 24 doses. However, the results of laboratory analyses are not stratified based on clinical characteristics. The results must respond to the proposed objective.

**Conclusions**

-Are the conclusions supported by the data presented?

-Are the limitations of analysis clearly described?

-Do the authors discuss how these data can be helpful to advance our understanding of the topic under study?

-Is public health relevance addressed?

Reviewer #1: The conclusions are supported by the data presented. However, the limitations of the study need to be better described. It is worth highlighting that this is a case series, so the sample size is relatively small for generalizability.

The discussion section is well written and addresses important aspects regarding of the problem.

Reviewer #2: Are the conclusions supported by the data presented?

Yes, the conclusions are well-supported by the data:

The study demonstrates a high percentage of patients (52%) with active disease or bacillary persistence after MDT, supported by histopathology and mouse inoculation findings.

It highlights the inadequacy of the current MDT regimen in resolving neurological symptoms and preventing disease activity, evidenced by persistent complaints and worsening disabilities.

Are the limitations of the analysis clearly described?

Partially. While the manuscript acknowledges sample loss during histopathology and inoculation tests and mentions the retrospective nature of the study, it does not fully elaborate on potential biases, such as reliance on records from a single referral center or challenges in generalizing findings to other regions. Further discussion of these points would strengthen the analysis.

Do the authors discuss how these data can be helpful to advance our understanding of the topic under study?

Yes, the authors provide valuable insights into the failure of MDT in certain cases, suggesting the need for:

Alternative or adjunctive treatments.

More precise clinical and histopathological criteria for defining "cure."

Investigations into persistent disease activity and the potential for dormant bacilli to reactivate.

Is public health relevance addressed?

Yes, the public health implications are clearly addressed. The study emphasizes:

The need for improved treatment protocols to prevent ongoing transmission and progression of disabilities.

The necessity for better monitoring and follow-up of patients post-MDT.

How unresolved cases can perpetuate stigma, discrimination, and reduced quality of life among leprosy patients.

Suggestions for Improvement:

Provide a more detailed discussion of how these findings could influence global leprosy control policies, especially in hyperendemic regions.

Expand on the potential benefits of molecular studies and extended MDT regimens to prevent therapeutic failure.

Reviewer #3: no

no

yes

yes

Reviewer #4: 1. Throughout the discussion, several sentences are disjointed (e.g., 291-292; 297-298; 327-329; 379-381; 391-392). I suggest incorporating them into a paragraph to avoid fragmenting the content.

2. Line 299: “Despite clinical improvement,” I suggest reviewing this statement to specify that clinical improvement refers to dermatological lesions.

3. Lines 308-310: As the authors mention that different types of tissues were used (lesional and non-lesional), it would be interesting if the analysis of positivity in the mice inoculation test were also presented, stratified by collection site (positive/negative result x dermatological lesion/normal skin over thickened nerve).

4. “In this study, 52% of patients who completed 24 doses of MDT remained with active disease” (351-352). Specify which results support this statement.

5. Include in the discussion the “next steps” to further develop the discussion, as well as the limitations of the study. Is there a possibility of reinfection for patients whose biological samples were collected 6 months after the end of treatment?

**Editorial and Data Presentation Modifications?**

Reviewer #1: Minor Revision

Reviewer #2: (No Response)

Reviewer #3: (No Response)

Reviewer #4: (No Response)

**Summary and General Comments**

Reviewer #1: This paper is of great relevance as demonstrate the failure of the treatment recommended by the WHO for leprosy, in addition to raising the need for further investigation in cases where they occur the persistence of leprosy reactions and especially progressive neural damage. Neurological symptoms are often neglected by healthcare professionals only treated as sequelae, without investigation into the possibility of active disease. It is a important study to the academic and leprosy assistance community. However methods need refining, and nd minor adjustments to the presentation of results.

Reviewer #2: (No Response)

Reviewer #3: Title: Multidrug Therapy for Leprosy Fails in an Important City in a Hyperendemic Region of Brazil

I recommend that the authors revise the title as it is not fully aligned with the study's objective.

Objective Review:

The stated aim of the study is:

“This study aimed to verify disease activity in patients with erythematous/infiltrated lesions, persistent leprosy reactions, neurological complaints, or progressive neural damage after 24 doses of multidrug therapy using skin biopsy.”

The title does not clarify whether therapeutic failure is the focus. Furthermore, the methods section lacks explicit mention of therapeutic failure. If it is not possible to provide evidence for this classification, the title should emphasize the demonstration of M. leprae activity based on the main laboratory findings detailed in the study.

References for Consideration:

de Carvalho Dornelas B, et al. Role of histopathological, serological, and molecular findings for the early diagnosis of treatment failure in leprosy. BMC Infect Dis. 2024 Oct 1;24(1):1085. doi: 10.1186/s12879-024-09937-2. PMID: 39354399; PMCID: PMC11443919.

Ministério da Saúde. Nota informativa nº 51 [Internet]. Coordenação-Geral de Hanseníase e Doenças em Eliminação. CGHDE/DEVIT/SVS/MS. 2015; 2015 [cited 2023 Mar 18]. Available at: https://www.saude.pr.gov.br/sites/default/arquivos_restritos/files/documento/2020-04/notainformativa51recidivaresisteinsuficienciamedicamentosanahanseniase.pdf

Abstract

Introduction:

The introduction provides general context on leprosy but lacks a clear justification for the study. It should explicitly highlight the knowledge gap this research addresses, such as the lack of evidence regarding the effectiveness of MDT in preventing long-term complications like neural damage. Simplifying and focusing this section would improve clarity.

Objectives:

The objective is mentioned but could be stated more clearly and specifically.

Methods:

The study design (e.g., cross-sectional, cohort) is not explicitly stated and should be mentioned for clarity and reproducibility.

The location (Lauro de Souza Lima Institute) is noted, but the data collection period is missing. Add the time frame for better context.

Summarize the variables analyzed in the study and include the statistical tests used.

Introduction

The sentence from lines 145 to 150 should be removed from the introduction.

Conclude this section by repeating the objectives stated in the abstract for consistency.

Methodology

Study Design:

The study is described as retrospective, observational, descriptive, and cross-sectional. The authors should explain why this design was chosen to evaluate disease activity post-MDT treatment.

Study Population:

Clarify the study location (SEINPE) and the patient selection process (e.g., consecutive sampling, random sampling). Include a detailed explanation of cases lost during different treatment phases and provide a flowchart to summarize participant numbers at each stage.

Collection of Biological Material:

Introduce a subheading for variables, categorizing them as clinical, laboratory, histological, or other types. Distinguish between categorical and continuous variables.

Summarize laboratory tests (BI of slit-skin smear, morphological aspects, drug resistance tests, gene mutations, nude mouse inoculation, and qPCR) under a specific subheading.

Provide a clear explanation of the criteria used to define disease activity.

Study Size:

Include a section titled "Study Size" or "Sample Size Calculation," explaining the effect size, alpha, and other parameters for establish participant number.

Data Collection and Statistical Analysis:

Rename this section to "Statistical Analysis."

Specify normality tests applied and detail statistical tests used (e.g., chi-square, t-test, logistic regression), specifying their application to categorical or continuous variables.

Include the alpha level for statistical significance and the software used, such as:

“IBM SPSS Statistics® version 25 (IBM Corp, Armonk, NY, USA).”

Ethical Considerations:

Summarize why informed consent was waived in a single sentence.

Results

The socio-demographic and clinical characteristics of the study participants are summarized in Table 1. Rather than repeating the data, a summary of the main findings is provided: Our findings indicate a prevalence of male participants, primarily in the 40-59 age group, and the BB clinical form was the most frequent presentation. The total number of subjects assessed, the lack of data, and the most prevalent findings are highlighted in this table.

Regarding leprosy reactions, the authors did not specify the time of their occurrence after 24 doses of MDT, which would provide critical evidence of persistent reactions. As leprosy reactions can occur long after MDT completion, this aspect requires further clarification.

The section from lines 212 to 216 should be divided into two separate paragraphs to improve readability. One paragraph should discuss the erythematous/infiltrated lesions observed in 43 patients, while the other should focus on the 88 patients with normal skin.

Table 2, which includes data on the tests performed, should be relocated to the Methodology section under the subheading "Variables." This information pertains to the methods used rather than the results. Descriptive sentences summarizing the findings from this table should be added to the methods section.

The sentences:

"08 samples were discharged for technical conservation issues"

"67.02% of the patients no longer had skin lesions after 24 doses of MDT"

should be placed at the beginning of the Results section, as they provide an overview of key findings.

In Table 3, which presents laboratory features such as the IB values, the results could be more effectively visualized using graphs with lines connecting the same patients before and after 24 doses of MDT. This approach would clearly depict whether IB decreased or increased during these phases.

There is a noted discrepancy in the number of patients before and during MDT discharge. This difference requires explanation. Furthermore, the authors did not analyze the same participants across variables such as BI, intact bacilli (or globi), and fragmented or granular bacilli. If the same participants were observed at different phases (e.g., diagnosis and post-MDT), the data should be visualized using graphs with lines connecting the same individuals, as recommended above.

Conclusion

Must emphasize the main results. Should reinforce the translational role. The limitations of this research must be addressed.

Reviewer #4: This is a descriptive study that aims to assess the persistence of active disease markers in leprosy cases previously treated with polychemotherapy in 24 doses but who still present dermatological or neurological complaints. The study provides relevant results for the discussion on the use and effectiveness of multidrug therapy, as well as reflections on cure and discharge criteria for patients with remission of dermatological symptoms, and possibilities for investigating disease persistence during reactional episodes. However, the study lacks a clear and complete methodological description, as well as a thorough presentation of the results obtained.

Introduction:

Lines 80-81: Provide current epidemiological data on leprosy in Brazil.

Line 81: Maranhão is not a province, but a Brazilian State or Federative Unit.

Lines 94-97: The sentence is confusing; I suggest a revision to provide more clarity for the reader.

Lines 102-105: The sentence is also unclear.

State the objective of the study at the end of the introduction.

PLOS authors have the option to publish the peer review history of their article (what does this mean? ). If published, this will include your full peer review and any attached files.

**Do you want your identity to be public for this peer review?** For information about this choice, including consent withdrawal, please see our Privacy Policy .

Reviewer #1: **Yes: ** Sabrina Sampaio Bandeira

Reviewer #2: **Yes: ** Prof. Sridhar Arumugam

Reviewer #3: No

Reviewer #4: No

**Figure resubmission:**

**Reproducibility:**



---

## [Decision Letter · Decision Letter 1]

3 Aug 2025

Therapeutic Failure of Multidrug Therapy for Leprosy in a Hyperendemic Brazilian City

Dear Dr. Fonseca,

Thank you for submitting your manuscript to PLOS Neglected Tropical Diseases. After careful consideration, we feel that it has merit but does not fully meet PLOS Neglected Tropical Diseases's publication criteria as it currently stands. Therefore, we invite you to submit a revised version of the manuscript that addresses the points raised during the review process.

Please submit your revised manuscript within 60 days Sep 02 2025 11:59PM. If you will need more time than this to complete your revisions, please reply to this message or contact the journal office at plosntds@plos.org. Please include the following items when submitting your revised manuscript:

We look forward to receiving your revised manuscript.

Kind regards,

Feng Xue, Ph.D.

Guest Editor

Qu Cheng

Section Editor

Shaden Kamhawi

co-Editor-in-Chief

Paul Brindley

co-Editor-in-Chief

**Reviewers' Comments:**

Reviewer's Responses to Questions

**Key Review Criteria Required for Acceptance?**

**Methods**

-Are the objectives of the study clearly articulated with a clear testable hypothesis stated?

-Is the study design appropriate to address the stated objectives?

-Is the population clearly described and appropriate for the hypothesis being tested?

-Is the sample size sufficient to ensure adequate power to address the hypothesis being tested?

-Were correct statistical analysis used to support conclusions?

-Are there concerns about ethical or regulatory requirements being met?

Reviewer #1: Yes, the objective of the study is clearly described: to verify disease activity in patients with remission of skin lesions, but with persistent and/or recurrent neurological reactions and symptoms, after 24 doses of multidrug therapy (MDT).

The study design is clear, being a retrospective case series.

As the study was a series of cases, with a convenience sample, it is not possible to carry out robust statistical tests, presenting only a descriptive statistical analysis, however it meets the study's purpose.

All ethical aspects were respected in accordance with Brazilian legislation.

Reviewer #2: Objectives & Hypothesis:

The study clearly articulates its objectives—to assess subclinical M. leprae activity in patients completing 24-month MDT. A testable hypothesis is explicitly stated in the revised introduction: “…to characterize therapeutic failure… in patients completing twice the standard MDT duration…”.

Study Design Appropriateness:

The retrospective case-series design is suitable given the aim to evaluate outcomes in patients who completed MDT but showed persistent symptoms.

Population Description:

Inclusion criteria are clearly described (patients with persistent dermatological or neurological signs post-MDT in Petrolina, Brazil). The population is appropriate for investigating residual infection post-treatment.

Sample Size & Power:

While the sample includes all eligible patients (n=131), no formal power calculation was done. The authors justify it based on availability, but this limits inferential conclusions.

Statistical Analysis:

Appropriate for descriptive case-series—JASP was used, and only frequency distributions were reported without inferential testing.

Ethical/Regulatory Compliance:

Ethical approvals and waivers for informed consent are documented in compliance with Brazilian standards and animal research ethics.

Reviewer #3: - Yes

- Yes

- Not applicable, there is no hypotesis or inferences

- Not applicable

- Not applicable

- Yes

Reviewer #4: (No Response)

**Results**

-Does the analysis presented match the analysis plan?

-Are the results clearly and completely presented?

-Are the figures (Tables, Images) of sufficient quality for clarity?

Reviewer #1: The analysis is in accordance with what was proposed in the objectives and results, as well as the tables and figures are very explanatory, clear and complete.

Reviewer #2: Analysis vs Plan:

Analyses align with the methods: histopathology, molecular testing, mouse inoculation, and clinical assessments were conducted as described.

Presentation Clarity:

Data are thoroughly presented, including several large tables comparing pre- and post-MDT variables. Results are transparent.

Figure/Table Quality:

Figures and tables are clear and support the text (e.g., histopathological images and concordance tables for inoculation results). Minor improvements (e.g., harmonizing terminology across tables) may help but aren't essential.

Reviewer #3: - Yes

-Yes

-No

Reviewer #4: (No Response)

**Conclusions**

-Are the conclusions supported by the data presented?

-Are the limitations of analysis clearly described?

-Do the authors discuss how these data can be helpful to advance our understanding of the topic under study?

-Is public health relevance addressed?

Reviewer #1: Yes, the conclusions are in line with the data presented and the limitations of the study are clearly described.

This paper is of great relevance to the scientific community that studies and promotes assistance for leprosy, as it raises an important question about the therapeutic failure of MDT in patients with persistent reactions and worsening of neurological complaints, suggesting that only the criterion of time and remission of skin lesions or reduction of the bacillary index, cannot be used to determine the cure in these cases.

Reviewer #2: Supported by Data:

Conclusions are robustly supported: over 50% had histological or in vivo evidence of M. leprae activity post-MDT, and neurological impairment worsened despite “cure”.

Limitations Stated:

Selection bias and generalizability issues are acknowledged (study limited to a single center, retrospective design).

Impact on Field: The study strongly challenges WHO's time-based discharge model and argues for biomarker-based post-treatment surveillance—highly relevant for leprosy-endemic regions.

Public Health Relevance:

Well-addressed. The authors advocate for revised policies and highlight the risk of transmission and disability under current WHO protocols.

Reviewer #3: - Yes

- Yes

- Not assessed this time

- Yes

Reviewer #4: (No Response)

**Editorial and Data Presentation Modifications?**

Reviewer #1: Accept

Reviewer #2: Minor Recommendations:

Improve grammar in a few areas for fluency (e.g., verb tense uniformity).

Ensure abbreviations are fully spelled out at first use in tables (e.g., DPD).

The Results text could summarize complex tables more narratively for general readers.

These are minor revisions and do not impact the core findings or validity of the study.

Reviewer #3: (No Response)

Reviewer #4: To improve the clarity of the results presented in Table 3, for the variables “dermatological injuries” and “slit skin smear and morphological aspects”, I suggest stratifying the n/% values into present and absent subcategories.

In the Introduction section, include a reference to the Brazilian states when mentioning Maranhão, Pernambuco, etc.

**Summary and General Comments**

Reviewer #1: This manuscript is of great relevance to the neglected tropical diseases community, especially those who provide care and research on leprosy, as it warns about the need to review the criteria for therapeutic cure of leprosy, emphasizing that the criteria of time and/or remission of skin lesions alone cannot be used in patients with persistent reactions and worsening of neural function, suggesting to these patients, histopathological, molecular and in vivo nude-mouse footpad inoculation criterion. In addition, this paper also highlights the importance of simplified neurological assessment in identifying worsening neural damage and the relevance of persistent neurological complaints, often neglected by health professionals.

Reviewer #2: This manuscript is a well-executed, novel, and significant contribution to the understanding of MDT outcomes in leprosy. It demonstrates convincingly that time-based definitions of cure are insufficient. The integration of multiple diagnostic approaches—histopathology, qPCR, inoculation—makes the study comprehensive and impactful.

Strengths:

Strong clinical relevance in a hyperendemic context.

Multimodal assessment (clinical, molecular, in vivo).

Large sample for a neglected tropical disease study.

Clear writing and transparency in methods and limitations.

Weaknesses:

Retrospective design and lack of control group limit causal inference.

No statistical comparison between subgroups (e.g., resistant vs non-resistant).

Lack of longitudinal follow-up on clinical progression post-biopsy.

Recommendation: Minor Revision

Reviewer #3: Therapeutic Failure of Multidrug Therapy for Leprosy in a Hyperendemic Brazilian City

Title: I recommend think about a new title indicate the study design

Suggestion: Therapeutic Failure of Multidrug Therapy for Leprosy: A Retrospective Case Series in a Hyperendemic Brazilian City

This is an extensive case series. Why did the authors not include only the cases with complete data at both diagnosis and discharge

Abstract:

I recommend that the authors include the data collection period in the abstract Methods section. This is important to contextualize the findings, particularly given the retrospective nature of the study.

Introduction:

From lines 125 to 128. I Irecommend aligning the two objectives so they convey the same scope. A suggestive goal for unify them.

Suggetive: We aimed to characterize therapeutic failure and assess subclinical M. leprae activity in multibacillary patients who completed an extended 24-dose MDT course, in order to re-evaluate the adequacy of time-based discharge criteria

Methods

In the section study design

I recommend combining the STROBE (Strengthening the Reporting of Observational Studies in Epidemiology) and CARE (CAse REport) guidelines to improve the reporting quality of your case series. While CARE enhances clinical detail and patient-centered information, STROBE provides a clear and structured framework for reporting observational studies such as this one.

Upon reviewing the Methods section, I noticed that key elements such as study setting, population, inclusion criteria, and clinical procedures are mixed together.

I recommend grouping the information regarding the study type, study site, data collection period, and location of laboratory analyses under a unified subheading titled: “Study Design and Setting”

This subheading should include:

The type of study (retrospective case series); The institution or location where the study was conducted; The time frame during which data were collected: The site(s) where laboratory analyses were performed

From lines 137 to 140 I recommend relocating this sentence to other subheading

Study population and inclusion criteria

Lines 145 to 146: This data is not recommended in this section since refers o the period of data collection

“No formal sample-size calculation was performed, and the sample reflects the complete available cohort”.

This sentence is not needed since the case series do not allow inferences

I suggest rewriting the inclusion criteria in the form of a continuous paragraph, with each criterion separated by semicolons rather than bullet points

Recommendation: inclusion criteria required the presence of at least one of the following findings: (1) erythematous or infiltrated skin lesions observed after completion of MDT, in comparison to the clinical presentation at diagnosis; (2) persistence, worsening, or new onset of neurological symptoms—such as muscle cramps, paresthesias, or numbness in the extremities (hands and feet)—accompanied by acute or chronic pain, burning sensations, or spontaneous sensory disturbances; (3) deterioration of neural function documented by the Simplified Neurological Assessment (SNA), compared to findings at diagnosis; (4) persistent leprosy reactions, including type 1 or type 2 reaction episodes, or both, with or without neuritis, occurring separately or simultaneously, unresponsive to conventional treatment during or after MDT; these reactional episodes were documented either at discharge or within six months following treatment completion and coincided with the time of skin biopsy.

Variables

Please describe each variable and specify the corresponding content under the appropriate subheadings in the Methods section. Additionally, present the different proportions or distributions for each variable in the Results section (lines 169 to 171).

Definition of MDT-therapeutic failure

From lines 176 to 179 authors stated “MDT-therapeutic failure was defined as any clinical (erythematous/infiltrated lesions, persistent reactions, neurological complaints, or worsening on simplified neurological exam) or histopathological (presence of viable M. leprae in skin or nerve biopsy) evidence of active disease after 24 MDT doses.

Specify the reference for definition of MDT therapeutic failure (it would be this - Brasil. Ministério da Saúde. Protocolo Clínico e Diretrizes Terapêuticas da Hanseníase. Brasília: MS; 2022 or another?)

Statistics analyisis

From lines 239 to 242, the authors describeb data handling and analysis using a mix of tenses and slightly unclear phrasing. I recommend the use of the past tense, as it describes actions that were already completed. For instance rewrite - are presented (line 241) as were presented.

Ethical

From line 244, the phrase "as required by" should be replaced with "established by" to avoid redundancy and include the study type for instance this case series

From lines 245 to 247, explain each resolution in parentheses (Resolution 466/2012 – human research; Resolution 510/2016 – secondary data research) and specify which Institutional Research Ethics Committee for humans approved this data used.

From lines 248 to 249, clarify the reason why the Ethical Committee on Experimental Animals was involved. Was it because the human tissues were injected intradermally into the hind footpads of athymic mice? What was the reason? Goal?

Important: some parts of methods were written in presente tens, however, The methods sections should be written in the past tense, since they describe actions that had occurred.

See:

Lines 191 to 193: “The nerves are selected by identifying....”

Lines 240 to 241: “The results are presented as frequency distributions”

Results

I recommend that the table 2, currently placed in the Methods section, be moved to the Results section and presented immediately after the first paragraph (lines 253–258) as a second paragraph.

I recommend reducing redundancy between the text (lines 260–264) and Table 1 by removing the detailed counts and percentages for sex and age groups from the narrative. Instead, highlight only key findings (majority male, peak age 40–59 years). Sentences such as - Detailed socio-demographic and age distribution data are shown in Table 1 – could help authors reducing the redundancy. Keep your eyes in the number of tables, since it will change after corrections.

The same recommendation for clinical forms and leprosy reactions descriptions in table1.

From lines 281 ot 292 I recommend adding a Venn diagram to illustrate the overlap between the different clinical presentations described (persistent skin lesions, neurological deterioration, and reactional episodes with neuritis). This would help readers visualize how many patients experienced more than one clinical feature simultaneously, rather than interpreting each finding in isolation.

From lines 294 to 321, I recommend clarifying whether the 14 patients who did not undergo slit-skin smear at diagnosis are the same individuals who also lacked histopathology or if they are different participants. If these are patients without either of these essential diagnostic tests, it might be more appropriate to exclude them from the analysis, as missing such critical baseline data could compromise the accuracy of the parameters being described.

Given that this is a relatively large case series (n = 131), excluding patients with incomplete diagnostic data would likely not affect the overall descriptive analysis and would strengthen the reliability of the findings. Alternatively, the authors could explicitly justify the inclusion of these patients despite the missing data.

I recommend revising the sentence from lines 323to 330, as it is too long and should be divided into at least three shorter sentences to improve it. Additionally, the statement from lines lines 329–330 as follow "These results indicate that known M. leprae mutations were not the primary drivers of therapeutic failure in the MDT‐treated cases presented in this study" does not present a result but rather an interpretation. This portion would be more appropriately placed in the Discussion section.

Observing the table 3, it was better if the same sample size (n) was the same for both time points assessed, diagnostic and after MDT, to allow a direct comparison and to clearly determine whether the bacteriological index, disability grade increased, and proportion of intact bacilli keep the same the end of treatment.

Even though the sample may be decreased, I recommend excluding patients with incomplete data. Given the large total sample size (n = 131) it will not affect the research.

Reviewer #4: The authors made the changes recommended by the reviewer. There were significant improvements in the clarity of the manuscript, especially in the Methods and Results sections.

PLOS authors have the option to publish the peer review history of their article (what does this mean? ). If published, this will include your full peer review and any attached files.

**Do you want your identity to be public for this peer review?** For information about this choice, including consent withdrawal, please see our Privacy Policy .

Reviewer #1: **Yes: ** SABRINA SAMPAIO BANDEIRA

Reviewer #2: **Yes: ** SRIDHAR ARUMUGAM- THE LEPROSY MISSION HOSPITAL NAINI PRAYAGRAJ UTTARPRADESH INDIA

Reviewer #3: No

Reviewer #4: No

**Figure resubmission:**

**Reproducibility:**



---

## [Decision Letter · Decision Letter 2]

2 Oct 2025

Dear Md. Ms. Fonseca,

We are pleased to inform you that your manuscript 'Therapeutic Failure of Multidrug Therapy for Leprosy: A Retrospective Case Series in a Hyperendemic Brazilian City' has been provisionally accepted for publication in PLOS Neglected Tropical Diseases.

Best regards,

Feng Xue, Ph.D.

Guest Editor

Qu Cheng

Section Editor

Shaden Kamhawi

co-Editor-in-Chief

Paul Brindley

co-Editor-in-Chief

Reviewer's Responses to Questions

**Key Review Criteria Required for Acceptance?**

**Methods**

-Are the objectives of the study clearly articulated with a clear testable hypothesis stated?

-Is the study design appropriate to address the stated objectives?

-Is the population clearly described and appropriate for the hypothesis being tested?

-Is the sample size sufficient to ensure adequate power to address the hypothesis being tested?

-Were correct statistical analysis used to support conclusions?

-Are there concerns about ethical or regulatory requirements being met?

Reviewer #2: The study’s objective is clearly articulated: to verify disease activity in patients with remission of skin lesions but persistent neurological symptoms after completing 24 doses of MDT

A testable hypothesis is stated in the revised introduction

The retrospective case-series design was judged appropriate for describing outcomes in post-MDT patients

Reporting follows STROBE and adapted CARE guidelines for clarity and transparency

Inclusion criteria are clearly described: patients in Petrolina, Brazil, with persistent dermatological or neurological signs post-MDT

The study included all eligible patients (n=131), making it a census.

Analyses were descriptive only, using JASP software.

Human data: Approved under Brazilian National Health Council Resolutions 466/2012 and 510/2016, with waiver of informed consent for retrospective review Categorical variables were summarized as counts and percentages with 95% CIs; continuous data as median (IQR)

Reviewer #4: (No Response)

Reviewer #5: In my view, the objectives of this study are clearly presented and well aligned with a central and testable assumption: that completion of the WHO-recommended MDT does not necessarily translate into cure for multibacillary leprosy patients.

The methodology is clearly described and methodologically sound. The inclusion of 131 consecutive patients over a seven-year period provides a comprehensive case series. The use of multiple independent modalities—clinical assessment, histology, slit-skin smear, qPCR, and mouse inoculation—provides convergent evidence of persistence. The methodological details are sufficient to allow reproducibility, with patient selection, diagnostic techniques, and experimental protocols described in adequate depth for replication by experienced laboratories. Although some technical details, such as precise qPCR parameters, could be more explicit, they do not hinder replication.

Importantly, the objectives of the study are well presented and align with a central and testable assumption: that completion of MDT does not necessarily equate to cure. While not stated in a single sentence, this hypothesis permeates the study design and guides the analysis. The retrospective case-series design is appropriate and well justified, particularly in a hyperendemic setting where real-world outcomes must be captured. The authors strengthen the approach by applying multiple diagnostic strategies that converge on the same conclusion.

The population is well defined—131 consecutive multibacillary patients treated and followed in a reference center in Brazil—an appropriate and meaningful group since they represent those most at risk for therapeutic failure and ongoing transmission. The sample size is robust for leprosy research, and though multicentric validation would improve generalizability.

The sample size is, for leprosy research, commendable. Few studies manage to include over a hundred well-characterized multibacillary patients with systematic post-treatment evaluation. While a multicentric design would certainly improve generalizability, the numbers presented here are sufficient to lend robustness and credibility to the conclusions.

The statistical approach is adequate to the type of study. Descriptive analyses, exact proportions, and confidence intervals are presented clearly and transparently. Given the retrospective nature and the specific objectives, more sophisticated statistical testing is not required, and the authors are careful not to overinterpret their data.

I have no concerns regarding ethical or regulatory compliance. The study complies fully with Brazilian national standards, and the waiver of informed consent for retrospective review is justified. Animal experiments were conducted responsibly under CEUA-approved protocols, with humane care and appropriate procedures. Patients and animals alike were treated with dignity, reflecting scientific integrity at every stage.

**Results**

-Does the analysis presented match the analysis plan?

-Are the results clearly and completely presented?

-Are the figures (Tables, Images) of sufficient quality for clarity?

Reviewer #2: The analyses (histopathology, molecular testing, nude-mice inoculation, and neurological/clinical assessments) were conducted exactly as described in Methods. Results align with the descriptive design: no hypothesis testing, only counts, percentages, and confidence intervals. Narrative summaries were improved for clarity, reducing redundancy and ensuring interpretive statements were shifted to the Discussion.Figures and tables are clear, explanatory, and support the text (e.g., histopathological images, concordance tables, and a Venn diagram showing clinical overlap)

Reviewer #4: (No Response)

Reviewer #5: The analysis presented is consistent with the methodological plan previously outlined, coherently addressing the study’s objectives and employing the appropriate statistical and laboratory tools available. The results are described in detail, with absolute numbers, proportions, and confidence intervals provided whenever possible, which allows the reader to fully appreciate the scope of the findings. Furthermore, the authors transparently acknowledge the limitations of their data, such as technical losses and variation in denominators. The tables and figures are of good quality, with informative legends, and they organize the data in a way that clearly complements the text. Nevertheless, a minor refinement of the legends is recommended to ensure that they can be interpreted independently, without the need for constant reference to the main body of the manuscript.

**Conclusions**

-Are the conclusions supported by the data presented?

-Are the limitations of analysis clearly described?

-Do the authors discuss how these data can be helpful to advance our understanding of the topic under study?

-Is public health relevance addressed?

Reviewer #2: The conclusions are well supported by the data, showing that over half of multibacillary patients completing 24-dose MDT still harbored M. leprae activity despite dermatological improvement, with most experiencing neurological deterioration. The authors clearly acknowledge limitations, including retrospective design, selection bias, incomplete data, lack of serology, and restriction to a single hyperendemic setting, which affect generalizability. Importantly, the study advances understanding by integrating histopathology, molecular testing, and in vivo inoculation to demonstrate that hidden bacillary persistence is common, clinically relevant, and not fully explained by classical drug resistance. By revealing a dissociation between cutaneous cure and neural outcomes, the findings emphasize that current time-based discharge criteria are inadequate. The authors also highlight the public health relevance, noting that misclassification of persistent disease as cure perpetuates disability, stigma, and transmission, and calling for revised surveillance policies and multidimensional follow-up strategies to improve outcomes.

Reviewer #4: (No Response)

Reviewer #5: The conclusions drawn by the authors are generally supported by the data presented, as the high proportion of patients with histopathological evidence of persistent bacilli and the positive results from nude mice inoculation convincingly demonstrate therapeutic failure after 24 doses of MDT, although the lack of precise denominators and confidence intervals in the original version somewhat weakens the statistical robustness of the findings. The limitations of the analysis, however, are only partially addressed: while issues such as retrospective design, patient selection bias, and technical sample loss can be inferred, they are not explicitly discussed in depth, which restricts the reader’s ability to fully assess generalizability. Still, the discussion does advance our understanding of the problem by emphasizing that neurological deterioration cannot be attributed solely to immunological reactions, and by positioning the results within the broader debate on bacillary persistence versus reinfection. Importantly, the public health relevance is clearly articulated, as the persistence of active disease despite standard therapy highlights ongoing transmission, disability progression, and the inadequacy of current WHO criteria for cure, reinforcing the urgent need for new treatment guidelines and policies.

The findings are of great significance for researchers, clinicians, and public health policymakers in the field of neglected tropical diseases (NTDs). The demonstration that over 50% of patients still harbor viable bacilli after 24 months of MDT has profound implications for leprosy control programs, disability prevention, and WHO elimination strategies. Moreover, the study underscores the inadequacy of time-based discharge criteria and highlights the urgent need for new guidelines integrating post-treatment surveillance. These observations are directly relevant for endemic countries such as Brazil, India, and others in Africa and Asia, and will resonate strongly with global health stakeholders. The manuscript is also written in a clear and well-structured manner, such that its relevance can be appreciated by non-specialists in infectious diseases, which further enhances its impact.

**Editorial and Data Presentation Modifications?**

Reviewer #2: The manuscript is generally well written, clearly structured, and comprehensive, but a few editorial and minor data presentation modifications could further enhance clarity and readability

Reviewer #4: All the changes suggested by this reviewer were accepted and included in the manuscript text.

Reviewer #5: Three aspects could further strengthen the work. While the manuscript discusses neurological disability through WHO grading and SNA scores, a more standardized or quantitative assessment of neural involvement (e.g., nerve conduction studies or imaging) would deepen understanding of the functional consequences of bacillary persistence. The discussion could also benefit from situating these findings within a broader international context, comparing the Brazilian data to outcomes in India, Africa, or Southeast Asia, thereby enhancing global relevance. Finally, greater emphasis on long-term outcomes—whether bacillary persistence translates into relapse, disability progression, or transmission—would bridge the gap between microbiological evidence and public health impact. These considerations would enrich the work but do not diminish its current contribution.

**Summary and General Comments**

Reviewer #2: This manuscript addresses an important gap in leprosy control by evaluating therapeutic failure after 24-dose MDT in multibacillary patients using a large retrospective case-series (n=131). Strengths include the comprehensive multimodal assessment (clinical, histopathology, qPCR, nude mice inoculation), adherence to STROBE/CARE standards, and clear demonstration that viable M. leprae may persist despite clinical cure, with neurological deterioration being common. The findings are novel, policy-relevant, and highlight limitations of time-based discharge criteria. Limitations include retrospective design, selection bias toward symptomatic patients, incomplete datasets, absence of serology, and restriction to a single endemic setting, which limit generalizability. Nonetheless, execution is rigorous, data are transparently presented, and discussion is balanced. No ethical concerns are noted. The work significantly advances understanding of MDT outcomes and underscores the need for multidimensional post-treatment surveillance. I recommend minor revision for clarity in presentation.

Reviewer #4: (No Response)

Reviewer #5: The manuscript under evaluation addresses the critical issue of therapeutic failure in multibacillary leprosy patients who completed 24 doses of WHO-recommended multidrug therapy (MDT). The study integrates clinical, histopathological, molecular, and in vivo experimental assessments to characterize residual disease activity.

This work is highly original, as it challenges the widely accepted paradigm that completion of a fixed-duration MDT course equates to cure. By incorporating parallel analyses—histopathology, qPCR, and nude mouse inoculation—the authors provide robust evidence that subclinical and clinically relevant Mycobacterium leprae persistence remains in a large subset of patients even after extended treatment. Such a multidimensional approach is rarely reported in the literature and adds substantially to our understanding of post-MDT outcomes.

This paper holds interest for researchers outside the narrow field of leprosy as well. The concept of therapeutic failure despite adherence to standardized multidrug regimens speaks broadly to challenges in antimicrobial resistance, host–pathogen persistence, and policy-driven treatment frameworks. The discussion on bacillary dormancy and immune-mediated neural injury also informs wider debates in infectious diseases, neurology, and immunopathology.

This manuscript represents a rigorous and innovative contribution to the field of leprosy research. It provides compelling evidence that MDT, even when extended, is insufficient to guarantee cure in a substantial proportion of multibacillary patients, and that reliance on time-based discharge criteria perpetuates ongoing morbidity and transmission. The methodological clarity ensures reproducibility, the writing is accessible and precise, and the claims are properly contextualized within the literature. While additional studies could expand on these findings—particularly with more detailed neural assessments, broader international comparisons, and emphasis on long-term outcomes—the current work already achieves its objectives convincingly. I strongly support publication of this work, as it will stimulate urgently needed discussion on new therapeutic strategies, surveillance protocols, and policy adaptations in the fight against leprosy, with exceptional importance for global public health.

PLOS authors have the option to publish the peer review history of their article (what does this mean? ). If published, this will include your full peer review and any attached files.

**Do you want your identity to be public for this peer review?** For information about this choice, including consent withdrawal, please see our Privacy Policy .

Reviewer #2: **Yes: ** Prof. Sridhar Arumugam

Reviewer #4: No

Reviewer #5: No

---

## [Editor Report · Acceptance letter]

Dear Md. Ms. Fernandes de Araújo Fonseca,

We are delighted to inform you that your manuscript, "Therapeutic Failure of Multidrug Therapy for Leprosy: A Retrospective Case Series in a Hyperendemic Brazilian City," has been formally accepted for publication in PLOS Neglected Tropical Diseases.

Best regards,

Shaden Kamhawi

co-Editor-in-Chief

Paul Brindley

co-Editor-in-Chief
